# Extreme optical nonlinearities unveiled by ultrafast laser filamentation in semiconductors

Maxime Chambonneau [1] ✉, Markus Blothe [1], Vladimir Yu. Fedorov [2], Isaure de Kernier [3], Stelios Tzortzakis [4,5] & Stefan Nolte [1,6]

Sky-high optical nonlinearities make semiconductors ideal platforms for multifunctional photonic devices. The fabrication of such complex devices could greatly benefit from in-volume ultrafast laser writing for monolithic and contactless integration. Ironically, as exemplified for Si, nonlinearities act as an efficient immune system that self-protects the material from internal permanent modifications. Predicting high-intensity ultrashort-pulse propagation beyond Si is further limited by incomplete descriptions of carrier dynamics in narrow-gap materials. Here, we demonstrate that filamentation universally dictates ultrashort laser pulse propagation in various semiconductors. The effective key nonlinear parameters extracted differ markedly from past measurements with low-intensity pulses, while temporal scaling laws for these parameters are also derived. Based on these findings, appropriate temporal-spectral shaping is proposed for tailored energy deposition inside semiconductors. The effective parameters also provide predictive inputs for semiconductor backside processing, microelectronics security, and high-harmonic, supercontinuum and terahertz wave generation.

Ultrafast laser filamentation is an extremely nonlinear propagation regime characterized by a dynamic balance between Kerr-induced self-focusing and plasma-induced defocusing[1,2]. In gases, the remarkable properties of filaments have led to a plethora of applications[3–9]. In wide-gap solids, the much higher nonlinear refractive index can be advantageously exploited for supercontinuum generation[10,11], laser direct writing of elongated structures[12] and fiber Bragg gratings[13]. However, in narrow-gap materials such as semiconductors, the understanding of ultrafast filamentation is, to date, limited to Si. In contrast with other media, nonlinear propagation effects in Si are disastrous when aiming for internal structuring. Strong low-order multi-photon absorption before the geometrical focus (i.e., prefocal absorption) coexists with filamentation and leads to delocalized energy deposition, which saturates below the modification threshold due to intensity clamping[14–20]. From previous studies of nonlinear propagation of femtosecond laser pulses in Si, various circumvention techniques have been devised for modifying the bulk of Si, by exploiting surface seeds[21,22], hyper numerical aperture[23,24], longer pulses in the picosecond[18,25–27] and nanosecond[28,29] regime, pulse trains[30], and mid-infrared pulses[31]. Nevertheless, the strong nonlinear absorption and the complex temporal electron dynamics observed during filamentation in Si, together with the limitations of nonlinear propagation models in accounting for these dynamics, prohibit generalization to other semiconductors. From the literature (Fig. 1), extreme nonlinear

[1]Friedrich Schiller University Jena, Institute of Applied Physics, Abbe Center of Photonics, Albert-Einstein-Straße 15, Jena, Germany. [2]Laboratoire Hubert Curien, Université Jean Monnet, Saint-Etienne, France. [3]First Light Imaging S.A.S., Europarc Ste Victoire Bât. 5, Route de Valbrillant, Meyreuil, France. [4]Institute of Electronic Structure and Laser (IESL), Foundation for Research and Technology–Hellas (FORTH), Heraklion, Greece. [5]Materials Science and Engineering Department, University of Crete, Heraklion, Greece. [6]Fraunhofer Institute for Applied Optics and Precision Engineering IOF, Center of Excellence in Photonics, Albert-Einstein-Straße 7, Jena, Germany. ✉e-mail: maxime.chambonneau@uni-jena.de

refraction is expected in these narrow-gap media. This suggests that filaments would form at modest laser pulse energies. Moreover, nonlinear absorption is exalted for narrow band gaps[32,33], which could indicate that, analogously to Si, prefocal absorption hinders localized energy deposition in the focal region.

In this article, we demonstrate that ultrafast laser filamentation dictates energy deposition in narrow-gap semiconductors. Two indirect (Si and Ge) and two direct (InP and GaAs) band-gap semiconductors have been selected (see the inset in Fig. 1). Besides technological importance due to their widespread use in microelectronics, photovoltaics, sensing, and quantum engineering, these materials exhibit cubic crystal structures, a property that minimizes anisotropy effects during nonlinear propagation. Because of their narrow band gaps ($E_g < 1.5$ eV), these media exhibit high linear and nonlinear refraction ($n_0 > 3$ and $n_2 > 10^{-18}$ m²/W, respectively). To work in their transparency spectral range, we employ ultrashort laser pulses at a wavelength of $\lambda = 1960$ nm (photon energy of 0.63 eV)[34]. This corresponds to the 2-photon absorption (2PA) regime for Si and Ge, and the 3-photon absorption (3PA) regime for InP and GaAs. We observe that filamentation governs the interaction in all tested semiconductors. We exploit the reduced energy deposition with ultrashort pulses below the modification threshold to determine the three-dimensional (3D) fluence distribution in these materials, which eventually leads us to define key nonlinear interaction parameters, including the peak fluence $F_p$, the effective critical power for nonlinearities $P_{cr}^{eff}$, and the effective 2- and 3-photon absorption coefficients $\beta_2^{eff}$ and $\beta_3^{eff}$. Repeating the measurements for pulse durations $\tau = 275$ fs – 25 ps, the temporal scaling laws for these parameters are determined. Ultimately, we propose temporal-spectral shaping approaches to increase energy deposition inside semiconductors.

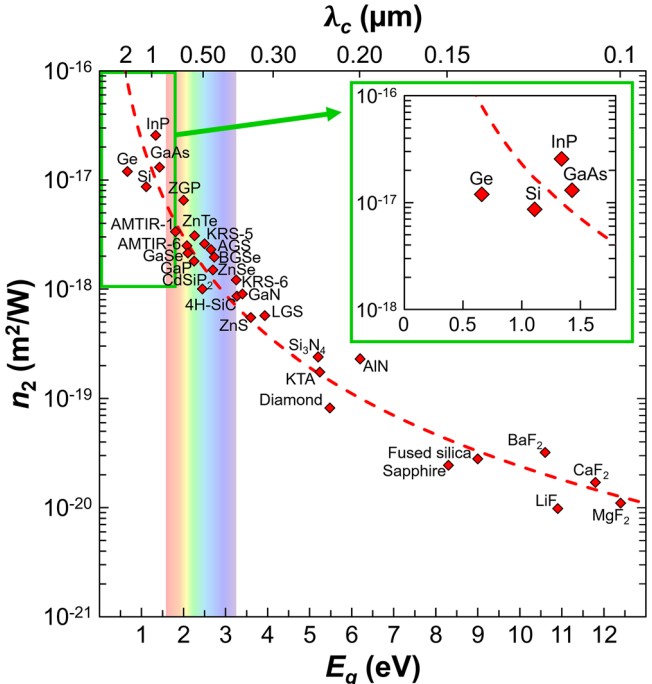

**Fig. 1 | Optical nonlinearities in various media according to the literature.** Dependence of the nonlinear refractive index $n_2$ of 30 materials on their band gap ($E_g$, corresponding to a cutoff wavelength $\lambda_c = hc/E_g$, where $h$ and $c$ are the Planck constant and the speed of light in vacuum, respectively). The data given at a wavelength of $\lambda = 1960$ nm or close to are extracted from the references indicated in Supplementary Note 1. The semiconductors explored in this study are highlighted in the inset.

## Results and Discussion

### A shared filamentation behavior

Our approach to characterize filamentation in semiconductors relies on nonlinear propagation imaging. This technique, analogous to tomography as illustrated in Fig. 2a, was initially developed for characterizing light propagation in water[35], and later applied to dielectrics[36,37]. It provides direct access to the laser energy per unit area [i.e., the fluence $F(x, y, z)$ in three spatial dimensions]. Concerning semiconductors, this technique was successfully employed to examine nonlinear propagation of light in Si[17,18,22,24,30,38], but it has been so far limited to this material. As shown by the fluence distributions in Fig. 2b, nonlinear propagation imaging is also applicable to other semiconductors. It is all the more favored by the fact that the modification threshold is not crossed (see Supplementary Note 2). For each material, increasing the input pulse energy $E_{in}$ qualitatively results in a spatially extended focal zone with respect to the linear regime. This is ascribable to the competition between Kerr and plasma effects, i.e., the formation of a filament. Quantitatively, the fluence distribution under identical laser conditions depends strongly on the medium's intrinsic nonlinear refraction and absorption properties. Nevertheless, a common feature between all materials is the $E_{in}$-dependent evolutive morphology of the fluence distribution, as exemplified in Fig. 2c. For low $E_{in}$ values, the propagation is linear, and the resulting fluence distribution takes the form of a *grain of rice*. The symmetry of this shape is broken for increased $E_{in}$, and distortions toward the prefocal region are observed. This *egg* morphology originates from the Kerr effect, which redistributes the fluence. When $E_{in}$ is further increased, prefocal absorption gives rise to the formation of wings, and an *angel* morphology appears. The angel wings form an angle which follows the half-angle of the cone of light $\theta = \arcsin(NA/n_0)$. This morphology eventually breaks up into multiple foci for the highest $E_{in}$. This *pearl necklace* morphology highlights the complex focusing and defocusing dynamics of filamentation. In addition, morphologies for high $E_{in}$ could be influenced by Fraunhofer diffraction patterns caused by the overfilling of the focusing optics. Notably, the position of the secondary on-axis lobes shifts nonlinearly with the input intensity (see Supplementary Note 4). The evolutive morphology in Fig. 2c has been observed for all tested semiconductors and pulse durations (see Supplementary Note 4).

Among all interaction parameters that can be extracted from the 3D fluence distributions (see hereafter), a low-hanging fruit is the maximum fluence $F_{max}$ defined as the highest value of the local fluence within the 3D spatial distribution [i.e., $F_{max} = \max(F(x, y, z))$]. As shown in Fig. 2d where $\tau = 900$ fs, in all semiconductors, $F_{max}$ scales linearly with $E_{in}$ for sub-nJ values. This is in excellent agreement with theoretical predictions in the linear regime (see Supplementary Note 3). In contrast, when $E_{in}$ exceeds a medium-dependent threshold, the experimental data deviate from this regime, and $F_{max}$ saturates to a peak value $F_p = \max(F_{max})$, which also depends on the material. This behavior is a direct consequence of intensity clamping, which is a typical feature of filamentation[1,2]. This implies that increasing the deposited energy in semiconductors by simply increasing $E_{in}$ is a strategy doomed to failure.

### Temporal scaling laws

Generally speaking, internal energy deposition strongly depends on the pulse duration, as the laser intensity scales inversely with this parameter. In contrast with wide-gap solids, it was demonstrated that longer pulses lead to higher intensities inside Si[18,25,26]. This counterintuitive result originates from decreased propagation nonlinearities when employing longer pulses, resulting in higher peak fluences. As shown in Fig. 3a, this trend is common to all examined semiconductors. Both the fluence saturation and the temporal dependence of $F_p$ support the conclusion that filamentation still dominates the interaction even for the longest pulses employed ($\tau = 25$ ps).

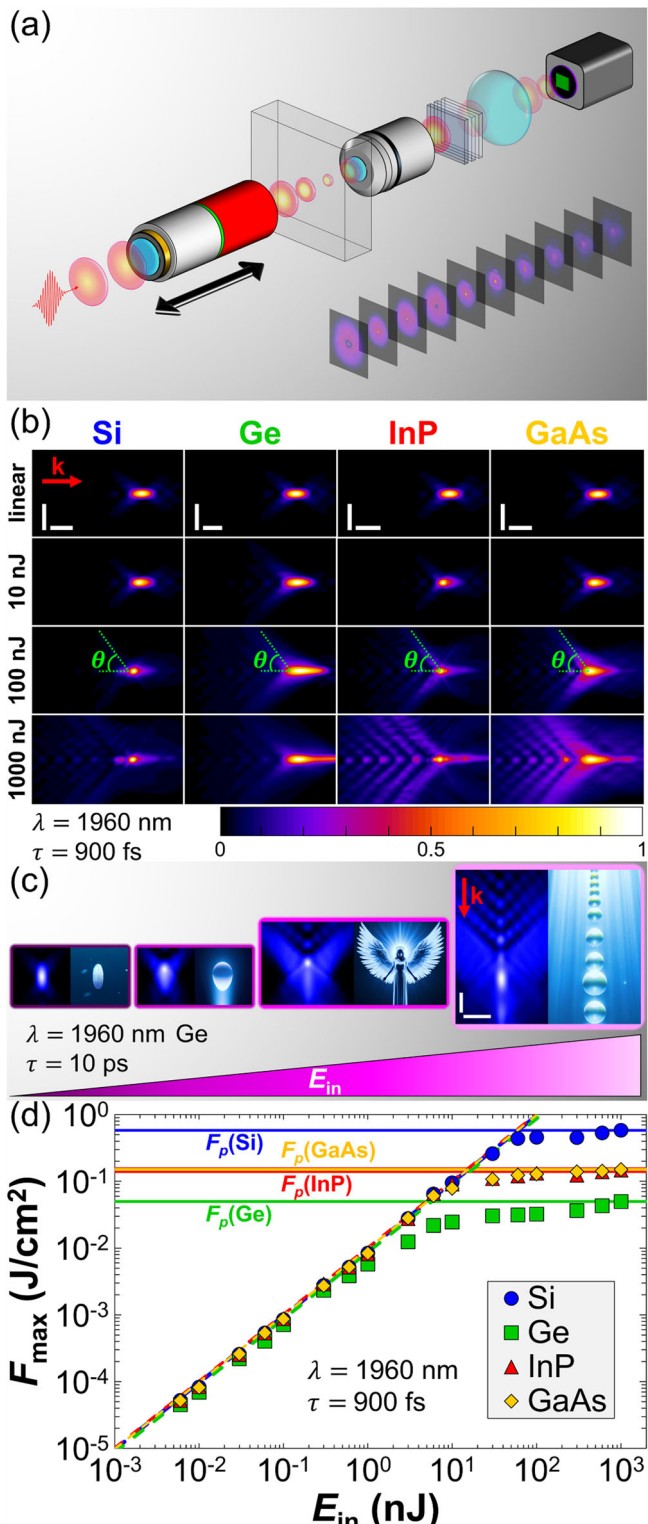

**Fig. 2 | Filamentation in different semiconductors. a** Schematic of the nonlinear propagation imaging set-up. **b** Normalized fluence distributions for various $E_{in}$ (pulse duration: $\tau = 900$ fs). The vector **k** indicates the direction of propagation, and $\theta = \arcsin(NA/n_0)$ indicates the medium-dependent half-angle of the cone of light. **c** Evolutive filament morphology with $E_{in}$, recorded in Ge ($\tau = 10$ ps). The displayed *grain of rice*, *egg*, *angel*, and *pearl necklace* morphologies are obtained for $E_{in} = 10$ pJ, 6 nJ, 60 nJ, and 600 nJ, respectively. **d** Evolution of the internal maximum fluence $F_{max}$ as a function of $E_{in}$ for different semiconductors ($\tau = 900$ fs). The dashed lines correspond to calculations in the linear propagation regime (see mathematical details in Supplementary Note 3). The radial and on-axis scale bars in (**b**) and (**c**) are 10 $\mu$m and 100 $\mu$m, respectively, and apply to all images for the same material.

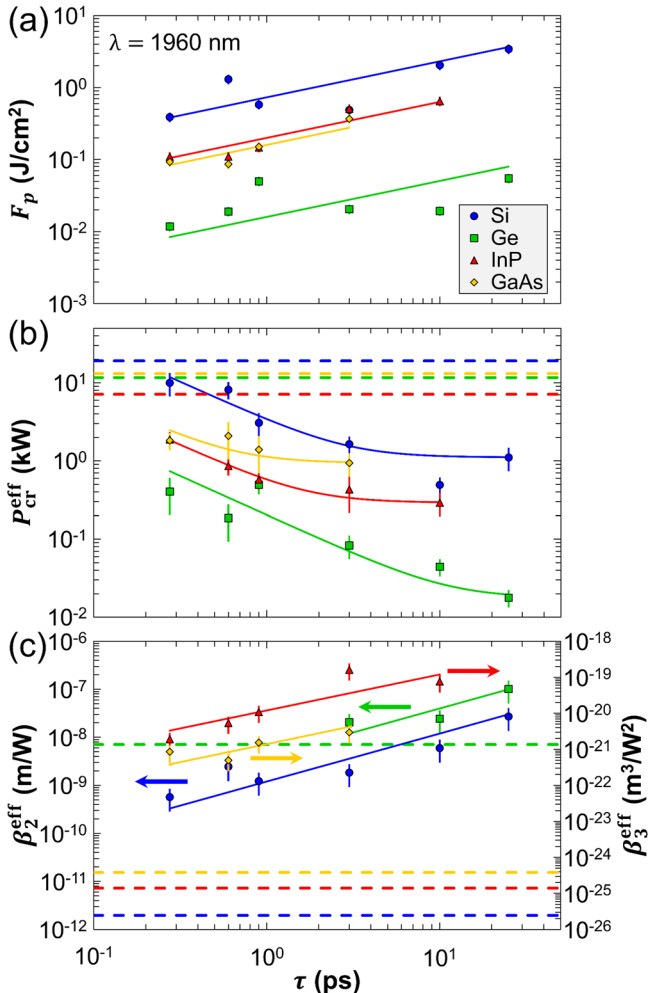

**Fig. 3 | Temporal scaling laws for key nonlinear coefficients in semiconductors.** Evolution of (**a**) the peak fluence $F_p$, (**b**) the effective critical power for nonlinearities $P_{cr}^{eff}$, and (**c**) the effective 2PA and 3PA coefficients $\beta_2^{eff}$ (for Si and Ge) and $\beta_3^{eff}$ (for InP and GaAs) as a function of the pulse duration $\tau$. The theoretical approach for determining $P_{cr}^{eff}$, $\beta_2^{eff}$, and $\beta_3^{eff}$ is given in Supplementary Note 4. The solid lines in (**a**) and (**c**) are $\sqrt{\tau}$ and linear fits, respectively. The solid lines in (**b**) are calculated from the model described in Supplementary Note 4, where a Gaussian medium response function is considered. The dashed lines in (**b**) and (**c**) correspond to literature values given in Supplementary Note 1. The error bars in (**a**–**c**) are defined in Supplementary Notes 2 and 4.

Interestingly, in the tested pulse duration range ($\tau = 275$ fs – 25 ps), the peak fluence $F_p$ scales as $\sqrt{\tau}$.

Besides the peak fluence $F_p$, we also determine key parameters related to nonlinear refraction and absorption. First, the experimental on-axis fluence profiles are compared to linear propagation calculations[39,40] to extract the effective critical power $P_{cr}^{eff}$ above which nonlinearities start to alter the propagation (see Supplementary Note 4 for more details on the method). While one could expect $P_{cr}^{eff}$ to be a constant material property, Fig. 3b shows that this parameter decreases with the pulse duration for all tested semiconductors. Such a temporal dependence of the effective critical power was observed for ultrashort pulse propagation in air[41,42] as well as fused silica[43], and explained by the fact that Kerr nonlinearity includes an instantaneous and a delayed medium response—the latter becoming non-negligible for longer pulse durations. Following an analogous approach for semiconductors, we demonstrate that the experimental trends in Fig. 3b may also originate from delayed medium response (Supplementary Note 4 for more details).

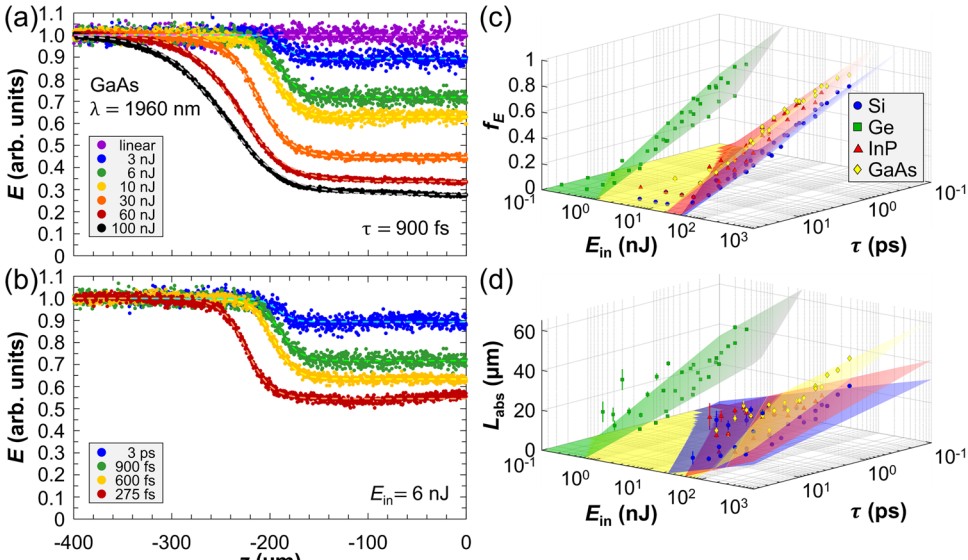

**Fig. 4 | Transverse space integration.** Examples of $E(z)$ profiles in GaAs are shown for various (**a**) input pulse energies with $\tau = 900$ fs, and (**b**) pulse durations with $E_{in} = 6$ nJ. Pulse duration and input pulse energy dependence of (**c**) the fraction of absorbed energy $f_E$, and (**d**) the characteristic absorption length $L_{abs}$ for all tested semiconductors. The data points are experimental values, and the corresponding surfaces are logarithmic fits, as detailed in Supplementary Note 4. The experimental error on $f_E$ in (**c**) is <10%. The error bars on $L_{abs}$ in (**d**) correspond to 95% confidence bounds on the sigmoid fit. Supplementary Videos 1 and 2 offer rotating visualization of the 3D plots in (**c**) and (**d**), respectively, and two-dimensional representations of these 3D plots are shown in Supplementary Note 4.

Nonlinear propagation imaging has also been exploited to extract the effective multi-photon absorption coefficient. To do so, the experimentally determined nonlinear focal shift is compared with our recent theoretical approach[38], which is based on a modified Marburger formula[44] that accounts for power losses. The effective critical power values $P_{cr}^{eff}$ in Fig. 3b are used as an input parameter for this model. While this approach was initially validated in the case of 2PA in Si, it is also applicable for all semiconductors in different multi-photon absorption regimes (Supplementary Note 4 for more details). The corresponding effective 2PA and 3PA coefficients $\beta_2^{eff}$ (for Si and Ge) and $\beta_3^{eff}$ (for InP and GaAs) both scale linearly with $\tau$ as shown in Fig. 3c.

An important result lies in the comparison between $P_{cr}^{eff}$, $\beta_2^{eff}$, $\beta_3^{eff}$, and the corresponding literature values for $P_{cr}$, $\beta_2$, and $\beta_3$ (see dashed lines in Fig. 3b, c). For both nonlinear refraction and absorption, the parameters deduced from nonlinear propagation imaging are orders of magnitude higher than the literature values. This can be ascribed to the methods traditionally employed for determining these nonlinear coefficients. For instance, in the most standard nonlinear optics technique (z-scan[45]), pulses with an energy right above the detection threshold for nonlinearities are loosely focused, which results in a change in transmission of a few percent. Therefore, the measured critical power and multi-photon absorption coefficient are valid for weakly excited media. In contrast, in our nonlinear propagation experiments with strongly ionized materials, the laser-produced plasma plays a critical role in nonlinear refraction (e.g., scattering, plasma defocusing) and absorption (e.g., free-carrier absorption, potentially leading to avalanche ionization).

Transverse integration of the fluence allows us to determine how the energy $E$ is absorbed during propagation along the optical axis $z$. Examples of normalized $E(z)$ profiles are shown in Fig. 4a, b for different $E_{in}$ and $\tau$ in GaAs. For intermediate $E_{in}$ values where the filament does not exhibit an *angel* or *pearl necklace* morphology, the experimental data are well-described by a sigmoid function (see Supplementary Note 4 for mathematical details). The sigmoid fits contain two key parameters. The first one is the fraction of absorbed energy $f_E$, which is determined by the ratio between the energies before and after the interaction. The second parameter is the characteristic absorption length $L_{abs}$, which is inversely proportional to the steepness of the sigmoid. Applying this fitting procedure for all conditions and materials, the dependence of $f_E$ and $L_{abs}$ on $E_{in}$ and $\tau$ (Fig. 4c, d, respectively) is obtained. Interestingly, $f_E$ and $L_{abs}$ both scale logarithmically with $E_{in}$ and also with $1/\tau$. These 3D plots show that high input intensity (i.e., high $E_{in}$ and short $\tau$) leads to a larger fraction of absorbed energy and an extended absorption zone along $z$. Combined with the saturation of the maximum fluence $F_{max}$ shown in Fig. 2d, this result shows that prefocal absorption governs the interaction in all tested semiconductors. Conversely, energy deposition is more localized for increased pulse durations.

## Temporal-spectral shaping

Besides this first solution, consisting of increasing the pulse duration $\tau$ to improve energy deposition in semiconductors, we have explored two additional methods, both relying on temporal-spectral shaping. The first method examined involves changing the temporal sequence of spectral components, i.e., the chirp. For the same duration, pulses that are not bandwidth-limited can be up- or down-chirped, i.e., the long wavelengths arrive first or last, respectively. While this parameter plays a major role in high-field physics[46], its effect is usually marginal at the moderate intensities typically involved in laser processing. Experimental studies on the surface of semiconductors showed that, in the linear absorption regime, the chirp has a moderate effect[47] or no effect at all[48] on the laser-produced carrier density. Analogous observations were made in the 2- and 5-photon absorption case in Si, where the plasma formation threshold is not significantly affected by the chirp[49].

In most of our experiments, the pulses are up-chirped. In Fig. 5a–c, the propagation of 3-ps pulses in Si is displayed for up- and down-chirped pulses. For the same $E_{in}$ value, the fluence distributions in Fig. 5a strongly differ between the two chirps. For down-chirped pulses, energy deposition is much more confined, as confirmed by the $F_{max}$ values in Fig. 5b for $E_{in} \geq 60$ nJ. For both chirps, a saturation plateau is reached for $E_{in} = 100$ nJ. However, for down-chirped pulses, $F_p$ is 2.4 × higher than for up-chirped pulses—thus demonstrating that down-chirped pulses are beneficial for improving internal energy deposition.

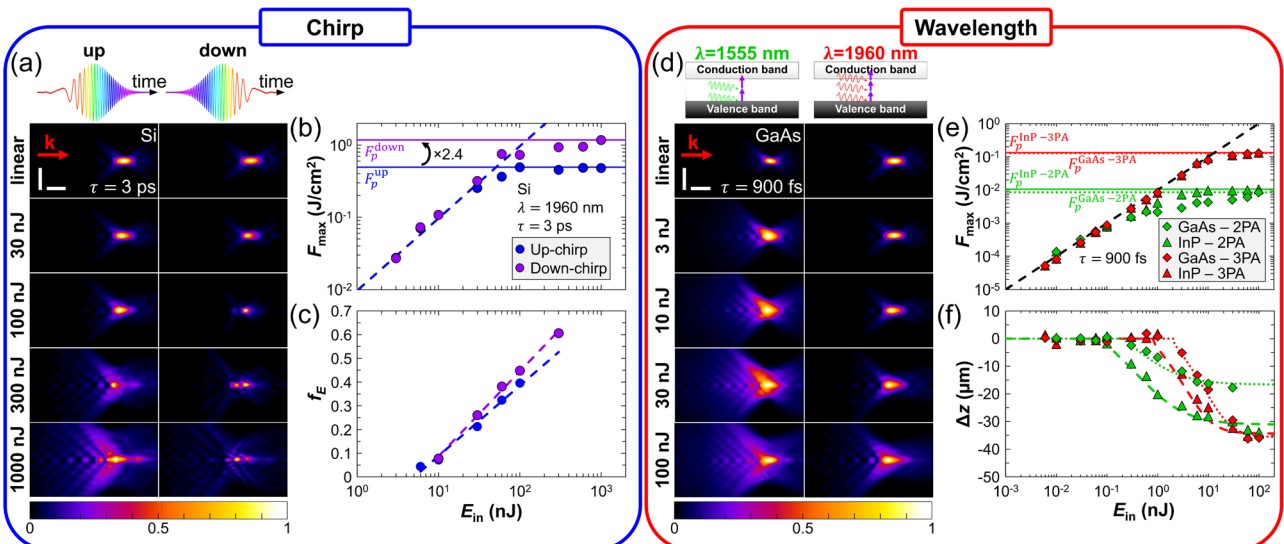

**Fig. 5 | Temporal-spectral shaping for optimizing energy deposition in semiconductors.** The temporal-spectral strategies consist of changing (**a**–**c**) the sign of the chirp, and (**d**–**f**) the multi-photon absorption order. **a** Normalized fluence distributions in Si for up- and down-chirped 3-ps pulses for various input pulse energies $E_{in}$. Evolution of (**b**) the maximum fluence $F_{max}$, and (**c**) the fraction of absorbed energy $f_E$ as a function of $E_{in}$ for both chirp configurations. The dashed lines in (**c**) correspond to logarithmic fits. **d** Normalized fluence distributions in GaAs for 1555-nm (2PA) and 1960-nm (3PA) 900-fs pulses for various input pulse energies $E_{in}$. Evolution of (**e**) the maximum fluence $F_{max}$, and (**f**) the nonlinear focal shift $\Delta z$ as a function of $E_{in}$ in InP and GaAs for both wavelengths. The dashed and dotted curves in (**f**) are calculated with the model described in Supplementary Note 4. The vertical and horizontal scale bars are 10 μm and 100 μm, respectively, and apply to all images in (**a**) and (**d**). The dashed lines in (**b**) and (**e**) correspond to calculations in the linear propagation regime (see mathematical details in Supplementary Note 3). The vector **k** indicates the direction of laser propagation.

This is all the more confirmed by the fraction of absorbed energy $f_E$ in Fig. 5c, which grows $\approx 20\%$ faster with $E_{in}$ for down-chirped pulses in comparison with up-chirped pulses, while the on-axis dimension of the interaction is similar for both chirp configurations (Fig. 5a). Employing the same analysis to determine the effective nonlinear coefficients as in Fig. 3, we find the same effective critical power $P_{cr}^{eff} = 1.6 \pm 0.4$ kW. However, the effective 2PA coefficient differs, with $\beta_2 = 1.8 \pm 0.92 \times 10^{-9}$ m/W and $3.0 \pm 1.5 \times 10^{-9}$ m/W for up- and down-chirped pulses, respectively, in agreement with the different nonlinear propagation behaviors in Fig. 5a.

Given that, for the measured smooth input spectrum, changing the chirp from up to down changes insignificantly the temporal intensity profile (see Supplementary Note 3), the higher peak fluence observed for down-chirped pulses can be explained by differences in ionization dynamics. Numerical studies for materials such as fused silica and $MgF_2$ [47,48,50] showed a similar asymmetry, as a consequence of multi-photon absorption being more efficient at shorter wavelengths—even for the same multi-photon order, as shown in Si [51,52]—whereas avalanche ionization becomes more efficient at longer wavelengths. The latter trend is directly described by the Drude model, which predicts that the inverse Bremsstrahlung heating rate scales as $\lambda^2$, so free carriers gain energy more efficiently from longer wavelengths. In the down-chirped configuration, the blue spectral components (shorter wavelengths) arrive earlier in the pulse, producing a higher initial free-carrier density that enhances avalanche ionization by the subsequent red components (longer wavelengths). As a secondary mechanism, plasma-induced anomalous dispersion may further strengthen this effect by causing the blue components to catch up with the red ones, compressing the pulse and increasing its peak intensity.

The second method, which has been explored to improve energy deposition in semiconductors, involves changing the multi-photon absorption order. This can be achieved by selecting different wavelengths corresponding to distinct multi-photon absorption regimes for the same medium. Establishing the optimal wavelength for which in-volume laser–semiconductor interaction is exalted is a problem with no intuitive solutions. Increasing the multi-photon absorption order would theoretically decrease energy deposition at the focus, but also limit prefocal absorption. Conversely, reducing the multi-photon absorption order would have the opposite effect. This problem was studied theoretically in Si, where it was predicted that the increased free-carrier absorption efficiency leads to higher energy deposition at longer wavelengths [16].

To experimentally evaluate the impact of multi-photon absorption order on filamentation, nonlinear propagation imaging has been performed in InP and GaAs for $\tau = 900$ fs at a wavelength of $\lambda = 1555$ nm (2PA), and then compared to the ones at $\lambda = 1960$ nm (3PA) in Fig. 5d–f. As exemplified for GaAs in Fig. 5d, the wavelength impacts the fluence distribution, both in the linear and the nonlinear regime. Prefocal absorption is much more pronounced for 2PA than for 3PA. Due to their direct band gap, 2PA is even more efficient in InP and GaAs at $\lambda = 1555$ nm compared to Ge at $\lambda = 1960$ nm (Fig. 2). For a given wavelength, an analogous evolution of $F_{max}$ for GaAs and InP is shown in Fig. 5e as a function of $E_{in}$. This suggests that filamentation governs the interaction both in the 2PA and the 3PA regime. However, an important result is that the peak fluence $F_p$ is more than one order of magnitude higher in the 3PA regime than in the 2PA regime, suggesting that higher multi-photon absorption orders are beneficial for tailoring energy deposition. This is consistent with results obtained in Si with 160-fs pulses in the 5-photon absorption regime [31], where permanent bulk modifications were achieved due to reduced prefocal absorption and the ability to reach the modification threshold. Although mid-infrared pulses generate lower electron densities, the deposited energy is increased through more efficient inverse Bremsstrahlung absorption [16]. These findings indicate that high-order multi-photon absorption can, under specific conditions, relax filamentation-induced constraints, whereas at shorter wavelengths filamentation still limits energy deposition in semiconductors.

Applying the same approach as for Fig. 3b, c, the nonlinear refraction and absorption parameters $P_{cr}^{eff}$ and $\beta_2^{eff}$ for $\lambda = 1555$ nm are determined. As shown in Fig. 5f, our theoretical approach successfully

reproduces the experimental evolution of the nonlinear focal shift $\Delta z$ as a function of $E_{in}$ for InP and GaAs at both wavelengths. For both materials, $P_{cr}^{eff}$ is one order of magnitude lower in the 2PA regime (57.1 ± 14.2 W for InP, 138.1 ± 69.0 W for GaAs) compared to the 3PA regime (573.7 ± 143.4 W for InP, and 1387 ± 693 W for GaAs). This is ascribable to the normal dispersion of the nonlinear index as well as the $\lambda^2$ dependence of the critical power. The data points where $P > P_{cr}^{eff}$ in Fig. 5f give access to $\beta_2^{eff} = 7.8 \times 10^{-8}$ m/W for InP, and $\beta_2^{eff} = 7.5 \times 10^{-8}$ m/W for GaAs. Similar to the measurement of $\beta_3^{eff}$ at $\lambda = 1960$ nm in Fig. 3c, these $\beta_2^{eff}$ values are orders of magnitude higher than the $\beta_2$ literature values[53–57], again attributable to differences in experimental determination method.

As a final remark, the three strategies to enhance energy deposition in semiconductors which consist of increasing $\tau$ (Fig. 3a), employing down-chirped pulses (Fig. 5b), and selecting $\lambda$ in higher multi-photon absorption order regimes (Fig. 5e) are not mutually exclusive. To illustrate this aspect, besides the narrow-gap semiconductors investigated in this study, it is interesting to note that the filamentation-caused limitations for energy deposition also persist in wider band-gap semiconductors such as polycrystalline ZnSe ($E_g \approx 2.7$ eV). Indeed, the threshold for internal modifications in this material could not be crossed with bandwidth-limited pulses with $\tau = 115$ fs at $\lambda = 800$ nm (2PA regime)[58]. In contrast, by simultaneously using longer pulse durations ($\tau = 0.5 - 2$ ps), down-chirped pulses, and $\lambda = 1047$ nm (3PA regime), in-volume laser writing was possible[59]. Apart from semiconductors, insulators exhibit even wider band gaps and lower nonlinear refractive index $n_2$ (Fig. 1). Consequently, the filamentation-caused limitations are drastically reduced in these materials, and internal modifications can be produced in the 2PA regime as exemplified for sapphire[60] and fused silica[61].

To summarize, nonlinear propagation imaging has demonstrated that filamentation governs ultrafast laser–matter interactions in semiconductors. The 3D fluence distributions obtained exhibit an evolutive morphology, as well as a saturation of the maximum fluence for increased input pulse energy. While, in principle, the impact of propagation nonlinearities would decrease with increasing pulse duration, the established temporal scaling laws for effective key nonlinear refraction and absorption parameters indicate that this strategy is less beneficial than expected. Nevertheless, for all tested semiconductors, the peak fluence increases with the pulse duration. Other identified strategies for increasing the peak fluence in semiconductors include temporal-spectral shaping, which can take the form of down-chirped or long-wavelength pulses. Among the tested optimization strategies, increasing the multi-photon absorption order by using longer wavelengths proved most effective for enhancing the peak fluence, followed by increasing the pulse duration, and finally by using down-chirped pulses. This hierarchy, quantified in the present work, provides practical guidance for tailoring ultrafast laser parameters in semiconductors. The determination of effective key nonlinear coefficients summarized in the Supplementary Note 4 is essential for predicting optimal conditions for ultrafast laser writing in semiconductors. Beyond this, these coefficients also provide critical insight into other fields, enabling advances for instance in backside processing, microelectronics security, THz wave generation, high-harmonic generation, and supercontinuum generation (see Supplementary Note 4).

## Methods
### Samples
The Si, Ge, InP, and GaAs samples used in this study are 500±25 $\mu$m thick, <100>-oriented, undoped, and two-sided polished.

### Ultrafast laser irradiation
A Tm-doped fiber laser source emitting at a center wavelength of 1960 nm and a repetition rate of 50 kHz has been used. The spectral bandwidth is 29 nm [full width at half maximum (FWHM)]. The pulse duration is measured with autocorrelation and is adjustable from 275 fs to 25 ps (FWHM assuming a sech$^2$ shape) by detuning the position of gratings in the optical compressor. In all experiments except the ones in Fig. 5a–c, the pulses are up-chirped, i.e., long wavelengths arrive before short wavelengths. The input pulse energy $E_{in}$ is measured at the sample position and controlled optically by means of a half-wave plate and a linear polarizer. The beam diameter before focusing is adjusted with a Galilean telescope to $\approx 12.7$ mm. The beam is focused by means of an objective lens of numerical aperture NA = 0.40 (Mitutoyo, M Plan Apo NIR 20×) mounted on a precision linear translation stage (Physik Instrumente, M-404.1DG) allowing its displacement along the optical axis with minimum incremental motion of 100 nm, and a resolution < 12 nm. For experiments at a wavelength of $\lambda = 1555$ nm, an Er-doped fiber laser source (Raydiance Inc., Smart Light 50) delivering non-chirped 900-fs pulses at a repetition rate of 1 kHz was employed. In all experiments the polarization is linear.

### Nonlinear propagation imaging
To record 3D fluence distributions inside semiconductors, nonlinear propagation imaging has been used. This technique detailed in Supplementary Note 2 is based on an infrared microscope directed opposite to the incoming laser. This microscope is composed of an objective lens of numerical aperture NA = 0.85 (Olympus, LCPLN100XIR), a tube lens (Thorlabs, TTL200-S8), and an extended InGaAs camera (First Light Imaging, C-RED 2 ER 2.2 $\mu$m). This infrared camera is composed of 640 × 512 pixels with 15 $\mu$m pixel pitch, and responds linearly in the 1380–2150 nm spectral range. For improved imaging performance, this camera has been operated with a frame rate of 600 Hz, an exposure time of 1.66 ms, and a sensor temperature of − 55°C. For experiments at a wavelength of $\lambda = 1555$ nm, a standard InGaAs camera (Xenics, Bobcat 320) composed of 320 × 256 pixels, with 20 μm pixel pitch, is used. To optimize the dynamic range of the cameras for all input pulse energies $E_{in}$ ranging from 1 pJ to 1 $\mu$J, calibrated neutral density filters have been inserted between the objective lens and the tube lens. The procedure can be divided into three steps. First, the focal plane of the imaging objective is positioned at the exit surface of the sample under white light illumination. Second, at low input pulse energy (typically, a few pJ), the geometrical focus is positioned at the exit surface of the sample, so that it is imaged on the camera. Third, the focusing objective is moved along the optical axis in 100 nm steps around the geometrical focus position for different input pulse energies. A typical recording consists of 2000 images, with a total stage movement of 200 μm around the geometric focus position. The recorded $z$ positions are then multiplied by the linear refractive index of the tested medium to get the actual positions in the material.

### Determination of the key nonlinear optical coefficients
To ensure that the 3D fluence distributions obtained for the lowest input pulse energies $E_{in}$ correspond to the linear propagation regime, propagation calculations detailed in Supplementary Note 3 have been performed with our vectorial model *InFocus*. In this regime, the maximum fluence reads $F_{max} = 2T_F E_{in}/(\pi w_0^2)$, where $T_F$ is the medium-dependent Fresnel transmission coefficient at the air–medium interface, and $w_0$ is the beam radius at $1/e^2$. The corresponding calculations serve as a benchmark for the evaluation of key interaction parameters such as the effective critical power for nonlinearities $P_{cr}^{eff}$ (see Supplementary Note 4), which is in turn implemented in our model based on a modified Marburger formula to determine the effective 2- and 3-photon absorption coefficients $\beta_2^{eff}$ and $\beta_3^{eff}$ (see Supplementary Note 4).

## Data availability
The data that support the results of this article are available within the Primary Manuscript or Supplementary Information.

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

## Acknowledgements

This research has received financial support from the German Federal Ministry of Research, Technology and Space [Bundesministerium für Forschung, Technologie und Raumfahrt (BMFTR)] through the RUBIN-UKPiño project (Grants no. 03RU2U033H, S.N., and 03RU2U032F, S.N.), the German Research Foundation [Deutsche Forschungsgemeinschaft (DFG)], through the Silabus (Grant no. 530105422, S.N.), and the Inseption (Grant no. 545531713, S.N.) projects, and the Hellenic Foundation for Research and Innovation (H.F.R.I.) under the "2nd Call for H.F.R.I. Research Projects to support Faculty members and Researchers" (Project Number: 4542, S.T.).

## Author contributions

M.C. and M.B. performed the experiments. M.B. automated the measurements. I.d.K. provided technical assistance with the sensor. M.C. processed, analyzed, and interpreted the data. V.Y.F., S.T., and M.C. developed the model and performed the numerical simulations. S.N. and M.C. led the project. All authors discussed the results and contributed to the manuscript prepared by M.C.

## Funding

## Competing interests

The authors declare no competing interests.
