## [Transparent Peer Review file · Nature Communications]

Extreme optical nonlinearities unveiled by ultrafast laser filamentation in semiconductors

Corresponding Author: Dr Maxime Chambonneau

Version 1:

Reviewer comments:

Reviewer #1

(Remarks to the Author)

This manuscript systematically investigates ultrashort laser pulse propagation in four kinds of narrow-gap semiconductors, and reveals the temporal scaling laws for key nonlinear parameters, which strongly differ from standard measurements with low-intensity pulses. Temporal-spectral shaping is finally proposed to optimize energy deposition inside semiconductors. These results lay the foundations for selectively tailoring semiconductors by ultrafast laser writing. Overall, the paper is interesting but raises some questions and demands several major improvements, which are listed below.

1. Two indirect (Si and Ge) and two direct (InP and GaAs) bandgap semiconductors have been investigated. It would be better to explain the reasons for choosing these specific semiconductors.
2. What is the specific definition of fluence? What the difference between the maximum fluence F_{max} and the peak fluence F_p ? Additionally, how to measure the F_{max} experimentally in this work?
3. How to obtain the normalized fluence distributions shown in Figure 2b based on the nonlinear propagation imaging in Figure 2a?
4. Figures 2d, 4band 4e demonstrate that the F_{max} vs. E_{in} satisfy power-law relationship. What is the fitted power exponent? Why?
5. Figure 4a demonstrates a significant difference in energy deposition between up- and down-chirped pulses at the same E_{in} at line 254. However, the subsequent analysis lacks a direct explanation for this phenomenon.
6. Regarding the explanation of Figure 4b, how to determine the factor of 2.4 in F_p between up- and down-chirped pulses at line 259?
7. What is the theoretical basis for the wavelength selection during the switching of multiphoton absorption orders at line 318?
8. The paper proposes three methods of optimizing energy deposition including increasing pulse duration, using a down-chirped pulse and achieving a higher multiphoton absorption order. Is there a hierarchy or relative priority among these methods? Furthermore, are there any quantitative metrics that can be used to evaluate the degree of energy deposition optimization?
9. The introduction inadequately covers the applications of ultrafast laser filamentation in gases.
10. In the Materials & methods, the section of propagation calculations mainly describes the calculation process of F_{max} , P_{cr}^{eff} , β_2^{eff} , β_3^{eff} , the title should be revised for clarity.
11. Besides ultrafast laser writing, supercontinuum generation during ultrashort laser pulses propagation in transparent solid is also widely concerned. Could the authors conduct a discussion regarding the impact of energy deposition on the supercontinuum generation?
12. Just like Figure 3b, please add the error bars and corresponding meanings for Figures 3a and 3c.

Reviewer #2

(Remarks to the Author)

In the manuscript "The universality of filamentation-caused challenges of ultrafast laser energy deposition in semiconductors", the authors analyze the influence of material properties, wavelength, pulse duration, and chirp of ultrashort laser pulses on the process of laser energy deposition in semiconductors. The manuscript has undeniable practical significance from the perspective of integrated 3D silicon photonics, as well as considerable fundamental importance, since

the interaction of ultrashort laser pulses with semiconductors remains still insufficiently studied. The work can be conditionally divided into several parts: the first investigates the influence of the material on the fluence profile within the volume of the semiconductor; the second discusses the impact of pulse duration and chirp on this distribution; and the last examines the influence of wavelength. The quality of the experimental and numerical modeling work is at a very high level. The literature review is sufficiently modern, and the results are novel. The work is potentially of interest to a broad readership.

Individually, the sections mentioned above are well-organized, have clear conclusions, and apart from some presentation features, I have practically no comments about them. However, combined, they do not form a cohesive single work, sometimes presenting too much information and raising questions that are not fully answered.

1) Firstly, at the beginning of the manuscript, in Fig. 1, the authors provide information on the bandgap, nonlinear and linear refractive indices, from which it follows that Ge differs most strongly from the other three samples, InP and GaAs have similar properties, and Si properties are somewhat intermediate (except for the nonlinear refractive index). However, later in the text, the maximum fluence is achieved precisely in silicon, and overall, the dependencies are most pronounced there (properties are least pronounced in germanium).

2) Secondly, the article varies several parameters simultaneously: wavelength, energy, duration, chirp, and material. As a result, after multiple readings, it creates an impression of some excessive clutter, and it is unclear which particular effect dominates in each case.

Based on these two points, at present, the open question remains: which specific material properties influence the propagation and absorption of ultrashort laser pulses in the mid-IR range in semiconductors? It seems there is no monotonic dependence on the bandgap, linear, or nonlinear refractive indices, although from general principles, one might expect that a larger bandgap and lower refractive indices (both linear and nonlinear) would lead to increased fluence. However, the data presented by the authors do not demonstrate this. Therefore, it might be advisable for the authors to consider a more comprehensive comparison of the effects of different regimes, perhaps in the form of one or several tables, or alternatively, splitting the work into multiple publications (even materials on silicon and/or germanium seem sufficient for a high-quality paper). As it stands, the number of samples rather confuses than adds value.

Besides these fundamental remarks, I note some less significant points:

3) I believe that the information about the influence of chirp is somewhat excessive for the main text of a data-heavy publication. Perhaps it could be moved to supplementary materials?

4) Fig. 3g,h are extremely poorly readable. Extracting information from them in their current form is very difficult.

5) Due to the large number of varying parameters and numerous figures, I recommend explicitly indicating the sample and the wavelength of the laser pulse on each figure. Working with the current presentation is quite challenging.

6) In Fig. 4d, the change in wavelength clearly affects the divergence angle of the "angel wings," see the figure below (in the pdf version of the review). Interestingly, in Figs. 4a, 2b, these angles are approximately the same. Is this caused by the change in the refractive index at the interfaces when changing the wavelength and, consequently, the effective focusing?

7) In Fig. 2b, there is a clear trend of increasing defocusing effect when varying the sample from Si → InP → GaAs → Ge. It might be worth changing the order to facilitate comparison of the impact regimes. Also, it seems that when changing the wavelength for the InP sample, the regime becomes similar to propagation in Ge. It would be advisable to align the energies presented in Figs. 2b, 4a, and 4c for better comparison.

8) The authors use the term "filamentation," but it seems that the length of the plasma channel is approximately the Rayleigh length, especially at low energies. Is it correct to use this term in this context?

In conclusion, I believe that this article undoubtedly deserves publication in Nature Communications, but only after major revision.

Reviewer #3

(Remarks to the Author)

The paper by Chambonneau et al. is aiming to provide a broad vision about physical origins of limitation conditions in semiconductors when using ultrafast laser pulses, and therefore explain the inability, in many cases, to induce modification inside such materials. It treats an important scientific and engineering problem for the development of 3D laser writing technologies applicable to semiconductors.

The manuscript is generally pleasant for reading, contains original data and is well-structured. The authors present interesting discussions based on observations of nonlinear propagation effects across variety of semiconductors. However, from both technical and fundamental viewpoints (explained below), the paper does not succeed to convincingly demonstrate truly "universal" filamentation process which would be responsible for the observed limitations. Although strong nonlinearities are indeed expected in the studied regimes, authors' choice to associate all observed effects under the same

term of “filamentation” is, in my opinion, often excessive and leads to confusion.

Even if this terminology might be justified depending on a chosen definition (which is not clearly given in manuscript, see after), the discussions do not systematically analyze the problem in view of competing effects trapping the light propagation as traditionally expected for filamentation phenomena (e.g. Kerr-based and plasma nonlinearities in the simplest case). Filamentation in usual sense implies intensity clamping as result. But intensity clamping itself can occur in absence of filamentation, and can originate from different nonlinear mechanisms (plasma mirror effect as an example in a different context).

Furthermore, paper does not bring new technique — it reproduces previous measurements done in silicon for other semiconductors. Also, no new physical concept or mechanism is directly demonstrated. Although the dependence on parameters like the chirp of the pulses is shown and is indeed interesting, authors themselves admit they do not provide a concrete solution or a clear new path to overcome the issue. They write that “the whole data set allows [them] to envision future optimizations”, but such statement stays vague and does not bring truly novel idea comparing to earlier reports.

Especially, I find not well addressed the question on high-order multiphoton absorption in comparison to a recent study on silicon using mid-IR femtosecond pulses (see ref.31). This previous study should be more critically discussed, because it suggests possibility for in-volume modification in silicon and may contradict the claim of an “universal” limitation due to filamentation. The current paper can be seen as directly supporting this alternative route, or contrarily as not fully compatible with the ‘universal’ picture stressed in it, making the delivered message unclear on this aspect.

My criticisms should not be understood as a questioning on the merit and interest of the work. The presented data are valuable and of interest for a specialized community. However, in my view, the paper does not offer sufficiently significant advancement in terms of new mechanism, concept, or technique to make it appropriate for a broad audience journal as Nature Communications.

Additional Comments:

- Positioning

- The general claim about broad potential impact of the findings is not clearly supported. The suggested advances in understanding and controlling nonlinear propagation regimes in semiconductors remain vague. Authors refer to applications in sensing, quantum engineering, healthcare, etc., but without concrete connection or perspective to actual implementations.

- I understand the motivation is to enable future modification inside semiconductors using ultrashort laser pulses. However, as stated in the introduction, such modifications have already been demonstrated in particular with longer and/or mid-IR pulses. Also, remarkable results exist using nanosecond pulses [refs 28, 29]. The paper does not explain sufficiently what specific new regime is targeted, and how it differs from these previous realizations. It does not describe what is exactly required to reach applications (e.g. specific material transformation) and how the current results are making critical step forward.

- Authors claim they identify filamentation regimes common to all semiconductors, beyond Si. But some statements are not accurate:

“However, in narrow-gap materials such as semiconductors, the understanding of ultrafast filamentation is to date limited to Si.”

This is misleading — literature reports filamentation also in SiC, GaAs, etc. (e.g. <https://doi.org/10.1016/j.ceramint.2024.02.133>, <https://doi.org/10.1007/BF00897938>). Possibly, authors mean understanding of filamentation in regard to the limitations for ultrafast laser 3D machining applications and add that “...nothing indicates that the conclusions on filamentation in Si hold in other semiconductors.” But even this statement appears questionable. After some literature check, I found no consensus that such limitations in Si are always due to filamentation. Intensity clamping is frequently observed, but is not necessarily result of filamentation process. Actually, as for dielectric studies the filamentation is rarely invoked in these works conducted with high NA focusing conditions. Also, somehow confirmed by Fig1, there is nothing which indicates a drastically different nonlinear propagation responses in other semiconductors than Si. The report is apparently a confirmation of this relatively intuitive aspect.

- The paper lacks a clear definition of “filamentation”. The link between observed intensity clamping and filamentation is often suggested but the direct relationship remains unclear (see previous comments). Statements like “prefocal absorption hinders localized energy deposition” or “prefocal absorption governs the interaction” appear contradictory with the concept of a situation fully governed by filamentation.

- Nonlinear regime categorization (Fig.2)

The proposed categorization of nonlinear propagation effects depending on laser power is interesting and represent the key contribution. However, the range of tested conditions seems not large enough to justify this claim of “universality”. The tested materials are relatively diverse, but most laser parameters are not varied significantly except pulse duration. In particular, it is

unclear how robust the observations are with respect to different focusing conditions and wavelengths. For example, in dielectrics, high-NA focusing usually hinders self-focusing and filamentation. In addition, some features observed (like “angel” or “pearl necklace”) could be influenced by the apparent Fraunhofer diffraction patterns in the focused beam, especially when beam largely overfills the focusing optics. These patterns have side lobes, secondary on axis maxima and structures that may play major role in studied cases. These aspects are not discussed, and robustness to an ideal Gaussian focusing for different NA is not demonstrated.

- Simulations and scaling laws

- Authors provide discussions based on pulse duration dependency. The scaling of saturation fluence $\propto \sqrt{t}$ is interesting, but used to claim that filamentation dominates even for long pulses (25 ps), which is unclear.

- Determination of effective nonlinear parameters (critical power, multiphoton coefficients, nonlinear refractive index) is valuable, but remains very indirect. In such complex beam propagation (see above comments on beam structures), rigorous 3D simulations accounting for nonlinear effects would be necessary to confirm the extracted values are meaningful. This would be important to confirm the effective parameters are not simply compensating other effects (not accounted) due to the simplifications of the treatments to make valid the coefficient determinations for the studied conditions. This makes very difficult the comparisons with well-known values measured rigorously for well-defined conditions with specific methodologies.

- Also, the methodology relies mostly on determinations of conditions for which deviations from linear propagation can be observed, but it is not strictly the critical power definition — it may for example detect early onset of detectable absorption, which is governed by multiple parameters not only the beam power.

The critical power dependence to the pulse duration is well supported by theoretical considerations in supplementary information about a nonlinear refractive index with a delayed response. While it is interesting and it sounds, without comparison to rigorous and complete propagation simulations I consider the authors should avoid to report on measurements on nonlinear coefficients in the studied conditions and try to compare to established values for different conditions. The employed methodology is highly performing for direct monitoring of intense beam behaviors and provides interesting observations. However, I do not consider it is the most appropriate methodology to determine nonlinear coefficients with precision.

- Potential temporal-spectral optimization

- The influence of pulse duration and wavelength is confirmed, but only two wavelengths are used (1.5 and 1.9 μm), which are relatively close. Thus, the conclusions about mid-IR and high-order multiphoton regimes are rather speculative.

- The sensitivity to chirp is an interesting observation. However, it seems to show a dependence, rather than demonstrate a major “control parameter”.

Moreover, with rather narrow spectrum (<30 nm), it is hard to attribute this to change in nonlinear effects without invoking resonance effects. Also, in experiments, any chirp change is usually associated with variation in pulse temporal shape and asymmetry, which can also affect results. Thus, definitive conclusions should be avoided unless complete temporal and spectral characterization is provided and/or results can be supported by rigorous simulations (see above).

- Additional minor comment

Fig. 1 compiles refractive index data from literature for many semiconductors and dielectrics which is surely interesting but appear not to be essential for the discussions of the paper.

Version 2:

Reviewer comments:

Reviewer #1

(Remarks to the Author)

The authors have addressed all issues that were pointed out in my review. Therefore, I recommend the paper for publication.

Reviewer #3

(Remarks to the Author)

As indicated in my original report, the paper by Chambonneau et al. addresses important scientific and engineering questions related to the development of new technologies based on in-volume interactions in semiconductors. In particular, the experimental work is carried out in configurations that are especially relevant to the challenge of 3D writing.

The revisions have produced a version that is convincing in some respects, notably through clearer definitions of the manipulated quantities (e.g., maximum fluence) and improved use of terminology. However, it remains less convincing with regard to the major conceptual concerns raised in the first round of review, as discussed below. Moreover, the additional material introduced in response to reviewers' comments, although informative in itself, tends at times to overload an already dense manuscript without significantly enhancing the clarity or impact of the conclusions.

Overall, I again recognize the substantial experimental effort and the merit of compiling such a large body of measurements. Nevertheless, I must reaffirm my initial impression: the work does not present a sufficiently novel conceptual contribution, nor does it offer a decisive advance in understanding or a clear technological direction. I therefore believe that the manuscript would be more appropriate for a more specialized journal.

In the present report, I do not revisit all points of disagreement with the authors but focus on a few key concerns that remain and that justify my overall assessment.

First, although I continue to question the appropriateness of grouping all observed effects under the term “filamentation,” I note and welcome the removal of the earlier claim of “universality,” which was not well supported and appeared overstated. Unfortunately, this change has been accompanied by the new claim of “Extreme optical nonlinearities,” which I find again somewhat artificial and even confusing once one considers how these values are obtained. The authors appropriately refer to “effective” nonlinear coefficients, but the extreme values they report arise primarily from the use of simplified retrieval models. While these effective coefficients are of interest for the particular experimental conditions studied here, they do not contradict the literature values obtained by standard methods and cannot be easily extrapolated to other conditions. Consequently, I do not consider it reasonable to use them to draw quantitative conclusions or make claims of broad applicability to different configurations such as looser focusing or waveguide-based applications (THz generation, high harmonics, supercontinuum, etc.).

Regarding the chirp dependence, I again acknowledge the interesting observation of a sensitivity to chirp, but I do not find in the added material more convincing evidence that chirp can be considered as a critical control parameter. In particular, the data do not demonstrate a sufficiently broad spectral coverage to fully support the discussions. Also, the observed symmetry of the temporal profiles appears to result from the reconstruction procedure itself, which retains only the quadratic phase term in Eq. S10 without measurement.

Finally, a major concern raised by reviewers in the first round was related to the absence of a meaningful quantitative metric to evaluate progress in addressing the challenges of 3D writing in semiconductors under nonlinear propagation effects. The metric introduced in the revision, the fluence F_p , seems to me entirely inappropriate. What matters here is the absorbed energy density (or maybe ionization level), not the maximum achievable fluence inside the material. To make this error obvious, one can consider the limiting case of a continuous, low-intensity radiation in the transparency domain of a semiconductor: in such a situation F_p can become arbitrarily large while the nonlinear absorption—and thus energy deposition—tends toward zero. This conceptual inconsistency undermines the reported ratios of improvement and the hierarchy of parameters (pp. 6–7). I would in fact strongly recommend suppressing these new sections, as they are misleading and add confusion rather than clarity.

Version 3:

Reviewer comments:

Reviewer #3

(Remarks to the Author)

As a preliminary remark, I observe that the authors' reply does not reassess the conceptual framing of the manuscript, nor does it provide additional experimental evidence that would reinforce the claim of wide applicability. The authors predominantly expand on their disagreements with the comments, yet do not seem to have considered a deeper or more comprehensive re-evaluation of the underlying issues.

Although the response concentrates on the few main points raised by previous report, none convincingly affects my overall evaluation. Again, my position does not concern the intrinsic interest of the study or the merit of the dataset, but reflects instead the high-level standards of originality and impact expected for this journal. From this perspective, the breadth of impact and the significance of the advances presented remain, in my view, insufficiently substantiated.

Regarding author response to R3 – Foreword

I appreciate the authors' efforts in providing a point-by-point reply. However, I note that the general concerns previously raised regarding conceptual novelty and/or understanding remain insufficiently addressed. The response focuses on individual remarks on the selection of deficiencies but does not directly engage with the broader issues which have been expressed.

Regarding author response to R3 – P1

I maintain that the Marburger formula, even when refined to account for nonlinear losses, while effective in describing the observations under the specific conditions of this study, cannot substitute to a rigorous nonlinear propagation model to support fully quantitative conclusions. Without a clear demonstration of the validity of the reported nonlinear coefficients in more general scenarios, they cannot be considered as contradicting others found in the literature.

Although the authors point out that supercontinuum generation is shown in S17, these observations remain relatively qualitative and do not directly validate the measured nonlinear coefficients. To substantiate claims of broad applicability, one would reasonably expect a range of different experiments (e.g., SC, THz, harmonic generation, material modification, ...) combined/compared with modeling that consistently supports the general relevance of the reported coefficients.

Regarding author response to R3 – P2

The authors' explanation regarding spectral dependencies offers some clarity that could improve the paper. Accordingly, I agree that even a relatively narrow spectrum may potentially lead to noticeable variations. However, the question of potential pulse asymmetry remains insufficiently addressed. While the quadratic phase term is indeed an appropriate simplification to describe the effect introduced by the stretcher, my initial remark was related to the spectral phase prior to the stretcher. Assuming an initially flat spectral phase leads naturally to a symmetric amplitude profile, but this assumption is not demonstrated. Figure R2-1 does not help to provide an unambiguous confirmation of symmetry in the conditions studied, and additional characterization would be required to fully support this point.

Regarding author response to R3 – P3

The response contains a certain inconsistency: the authors state that the relevance of the chosen parameters extends well beyond 3D writing, yet they justify their choice primarily by referring to studies on 3D writing or material modification. While I agree that using a directly measurable metric is advantageous, my concerns remain regarding its general relevance for a broad range of nonlinear optics experiments.

The issue I initially raised with a metric that can become arbitrarily large even in a regime insufficient to trigger any nonlinear response has not been convincingly addressed. Referring to works involving CW-induced modifications does not clarify how the proposed metric can be consistently used to identify an optimum in the present context. Therefore, I must respectfully but firmly disagree with the authors' assertion that my earlier remarks were "incorrect" and I must maintain my position regarding the limited relevance of the reported ratios.

Reviewer #4

(Remarks to the Author)

Response to Reviewers

Contents

Response to Reviewer #1	2
Reviewer #1, Foreword	2
Reviewer #1, Point 1	2
Reviewer #1, Point 2	2
Reviewer #1, Point 3	3
Reviewer #1, Point 4	4
Reviewer #1, Point 5	4
Reviewer #1, Point 6	5
Reviewer #1, Point 7	5
Reviewer #1, Point 8	5
Reviewer #1, Point 9	6
Reviewer #1, Point 10	6
Reviewer #1, Point 11	7
Reviewer #1, Point 12	7
Response to Reviewer #2	9
Reviewer #2, Foreword	9
Reviewer #2, Point 1	9
Reviewer #2, Point 2	9
Reviewer #2, Point 3	10
Reviewer #2, Point 4	11
Reviewer #2, Point 5	11
Reviewer #2, Point 6	11
Reviewer #2, Point 7	12
Reviewer #2, Point 8	12
Reviewer #2, Conclusion	13
Response to Reviewer #3	14
Reviewer #3, Foreword	14
Reviewer #3, Point 1	14
Reviewer #3, Point 2	14
Reviewer #3, Point 3	15
Reviewer #3, Point 4	16
Reviewer #3, Point 5	16
Reviewer #3, Point 6	17
Reviewer #3, Point 7	18
Reviewer #3, Point 8	19
Reviewer #3, Point 9	19
Reviewer #3, Point 10	20
Reviewer #3, Point 11	21
Reviewer #3, Point 12	22
Reviewer #3, Point 13	23
Reviewer #3, Point 14	23
Reviewer #3, Point 15	25
References	26

Response to Reviewer #1

Reviewer #1, Foreword

Reviewer #1: This manuscript systematically investigates ultrashort laser pulse propagation in four kinds of narrow-gap semiconductors, and reveals the temporal scaling laws for key nonlinear parameters, which strongly differ from standard measurements with low-intensity pulses. Temporal-spectral shaping is finally proposed to optimize energy deposition inside semiconductors. These results lay the foundations for selectively tailoring semiconductors by ultrafast laser writing. Overall, the paper is interesting but raises some questions and demands several major improvements, which are listed below.

Authors: We sincerely thank Reviewer #1 for the thorough and constructive review. We greatly appreciate the positive assessment of our work, particularly regarding the systematic investigation of ultrashort laser pulse propagation in four narrow-gap semiconductors and the potential of temporal-spectral shaping to control energy deposition. The insightful comments have been invaluable in improving the clarity and rigor of the revised manuscript. We address each concern point by point below. For clarity, reviewer comments are shown in **blue**, our responses in **green**, and modifications in the revised manuscript in **red**.

Reviewer #1, Point 1

Reviewer #1: Two indirect (Si and Ge) and two direct (InP and GaAs) bandgap semiconductors have been investigated. It would be better to explain the reasons for choosing these specific semiconductors.

Authors: Beyond demonstrating that filamentation-induced limitations in Si generalize to other semiconductors, we selected these four materials for two main reasons. First, they are all technologically important and widely used in diverse applications, such as microelectronics, photovoltaics, sensors, and quantum engineering. Second, all these narrow-gap materials are transparent at a wavelength of $\lambda = 1960$ nm. At this wavelength different material-depended multi-photon absorption orders dominate (2-photon absorption for Si and Ge, and 3-photon absorption for InP and GaAs). Finally, all exhibit cubic crystal structures (diamond cubic for Si and Ge; zinc blende for InP and GaAs), which is a key feature that reduces anisotropy and polarization dependence during nonlinear propagation.

Revisions brought to the manuscript: To reflect both points, the original statement:

“Two indirect (Si and Ge) and two direct (InP and GaAs) band-gap semiconductors have been selected (see the inset in Fig. 1). Because of their narrow band gaps ($E_g < 1.5$ eV), these media exhibit high linear and nonlinear refraction ($n_0 > 3$, and $n_2 > 10^{-18}$ m²/W, respectively).”

has been revised to:

“Two indirect (Si and Ge) and two direct (InP and GaAs) band-gap semiconductors have been selected (see the inset in Fig. 1). Besides technological importance due to their widespread use in microelectronics, photovoltaics, sensing, and quantum engineering, these materials exhibit cubic crystal structures, a property that minimizes anisotropy effects during nonlinear propagation. Because of their narrow band gaps ($E_g < 1.5$ eV), these media exhibit high linear and nonlinear refraction ($n_0 > 3$ and $n_2 > 10^{-18}$ m²/W, respectively).” (Primary Manuscript, page 2)

Reviewer #1, Point 2

Reviewer #1: What is the specific definition of fluence? What the difference between the maximum fluence F_{\max} and the peak fluence F_p ? Additionally, how to measure the F_{\max} experimentally in this work?

Authors: The reviewer is right, and we agree that defining terms is essential for the clarity of our manuscript. The fluence $F(x, y, z)$ is defined as the laser energy per unit area. The maximum fluence F_{\max} is the highest fluence over all three spatial dimensions, and reads

$$F_{\max} = \max_{x,y,z} (F(x, y, z)). \quad (\text{R1})$$

For linear propagation, F_{\max} is proportional to the input pulse energy E_{in} . However, in the nonlinear propagation regime, as F_{\max} saturates for increased pulse energy E_{in} [see the plateaus in Fig. 2(d)], the peak fluence F_p is introduced as the highest F_{\max} value among all input pulse energies E_{in} , and reads

$$F_p = \max_{E_{\text{in}}} (F_{\max}) = \max_{x,y,z,E_{\text{in}}} (F(x,y,z)). \quad (\text{R2})$$

Revisions brought to the manuscript: To clarify these definitions, the original statement:

“This technique, analogous to tomography as illustrated in Fig. 2(a), was initially developed for characterizing light propagation in water [34], and later applied to dielectrics [35, 36]. Concerning semiconductors, it was successfully employed to examine nonlinear propagation of light in Si [17, 18, 22, 24, 30, 37], but it has been so far limited to this material.”

has been revised to:

“This technique, analogous to tomography as illustrated in Fig. 2(a), was initially developed for characterizing light propagation in water [34], and later applied to dielectrics [35, 36]. *It provides direct access to the laser energy per unit area [i.e., the fluence $F(x,y,z)$] in three spatial dimensions.* Concerning semiconductors, *this technique* was successfully employed to examine nonlinear propagation of light in Si [17, 18, 22, 24, 30, 37], but it has been so far limited to this material.” (Primary Manuscript, page 2)

The original statement:

“Among all interaction parameters that can be extracted from the 3D fluence distributions (see hereafter), a low-hanging fruit is the maximum fluence F_{\max} .”

has been revised to:

“Among all interaction parameters that can be extracted from the 3D fluence distributions (see hereafter), a low-hanging fruit is the maximum fluence F_{\max} *defined as the highest value of the local fluence within the 3D spatial distribution [i.e., $F_{\max} = \max_{x,y,z} (F(x,y,z))$].*” (Primary Manuscript, pages 2 and 3)

The original statement:

“In contrast, when E_{in} exceeds a medium-dependent threshold, the experimental data deviate from this regime, and F_{\max} saturates to a peak value F_p , which also depends on the material.”

has been revised to:

“In contrast, when E_{in} exceeds a medium-dependent threshold, the experimental data deviate from this regime, and F_{\max} saturates to a peak value $F_p = \max_{E_{\text{in}}} (F_{\max})$, which also depends on the material.” (Primary Manuscript, page 3)

Reviewer #1, Point 3

Reviewer #1: How to obtain the normalized fluence distributions shown in Figure 2b based on the nonlinear propagation imaging in Figure 2a?

Authors: The fluence distributions in Fig. 2(b) are reconstructed from a stack of xy images recorded at successive on-axis positions z using the nonlinear propagation imaging setup shown in Fig. 2(a). For each z -position, the transmitted light intensity map is recorded, background-subtracted, and converted into absolute fluence using calibration with a reference measurement of the pulse energy. The resulting 3D fluence distribution $F(x,y,z)$ is then normalized at each z -plane to its maximum value $F_{\max}(z)$ to emphasize the relative spatial distribution, independent of energy fluctuations. These normalized slices are assembled to produce the plots in Fig. 2(b). Detailed image processing steps and the normalization procedure are provided in the Supplementary Information (Section S2.1), along with a schematic workflow in new Fig. S1.

Revisions brought to the manuscript: To clarify how to obtain the fluence distributions from nonlinear propagation imaging, and to provide more technical details concerning the image processing, we have added a dedicated section in the Supplementary Information (S2.1 **Nonlinear propagation imaging technique**), which includes a new figure (Fig. S1).

The original statement:

“This technique is based on an infrared microscope directed opposite to the incoming laser.”

has been revised to:

“This technique *detailed in Supplementary Information, Section S2.1* is based on an infrared microscope directed

Reviewer #1, Point 4

Reviewer #1: Figures 2d, 4b and 4e demonstrate that the F_{\max} vs. E_{in} satisfy power-law relationship. What is the fitted power exponent? Why?

Authors: Actually, in these figures [now, Figs. 2(d), 5(b) and 5(e) in the revised manuscript], the saturation for high input pulse energy values implies that a power-law function—or, more generally, any mathematical function which tends to infinity when $E_{\text{in}} \rightarrow +\infty$ —will fail to fit the whole set of F_{\max} data. Nevertheless, in the linear propagation regime (i.e., for low E_{in} values), F_{\max} scales linearly with E_{in} , which can be viewed as a power-law relationship with an exponent of 1. This is highlighted by the dashed lines in these figures, which do not result from a fitting procedure, but instead come from calculations detailed in Supplementary Information, Section S3.2, where we show for a Gaussian fluence distribution that

$$F_{\max} = \frac{2T_F E_{\text{in}}}{\pi w_0^2}, \quad (\text{R3})$$

where $T_F = 1 - (1 - n_0)^2 / (1 + n_0)^2$ is the Fresnel transmission coefficient at the air–medium interface at normal incidence, n_0 is the linear refractive index of the material, and w_0 is the beam radius at $1/e^2$.

Revisions brought to the manuscript: To clarify this aspect, the original statement:

“The dashed lines correspond to the linear propagation regime.”

has been revised to:

“The dashed lines correspond to *calculations in the linear propagation regime (see mathematical details in Supplementary Information, Section S3).*” (Primary Manuscript, Fig. 2 caption)

The following statement has been added to the caption of the new Fig. 5:

“*The dashed lines in (b) and (e) correspond to calculations in the linear propagation regime (see mathematical details in Supplementary Information, Section S3).*” (Primary Manuscript, Fig. 5 caption)

Reviewer #1, Point 5

Reviewer #1: Figure 4a demonstrates a significant difference in energy deposition between up- and down-chirped pulses at the same E_{in} at line 254. However, the subsequent analysis lacks a direct explanation for this phenomenon.

Authors: We agree with Reviewer #1, and we have added a clearer and more explicit explanation for the chirp dependence. This explanation, which also addresses Reviewer #3, Point 14, relies on the fact that the measured temporal profiles of up- and down-chirped pulses are almost identical, highlighting that the observed differences in detected fluence are due to the different arrival sequence of the spectral components rather than pulse-shape distortions.

Revisions brought to the manuscript: To clarify this aspect, the original statement:

“The chirp-dependence of filamentation may originate mainly from two physical phenomena. First, the ionization dynamics could differ between up- and down-chirped pulse. Calculations showed that a similar asymmetry exists for fused silica and MgF_2 [44, 45, 47]. This is ascribable to the increased efficiency of multi-photon absorption for shorter wavelengths (even for the same multi-photon absorption order [48, 49]), while avalanche ionization is more efficient for longer wavelengths. Therefore, in the down-chirped configuration where the “blue” spectral components arrive before the “red” ones, the produced free-carrier density is higher. The second phenomenon which could play a role is nonlinear dispersion induced by the plasma. Indeed, the interaction results in extended plasma channels along the optical axis, which may show anomalous dispersion, meaning that the “blue” can catch up with the “red” part of the pulse, in turn resulting in shorter duration—and thus, higher intensity.”

has been revised to:

“*Given that, for the measured smooth input spectrum, changing the chirp from up to down changes insignificantly the temporal intensity profile (see Supplementary Information, Section S3.4), the higher peak fluence observed for down-chirped pulses can be explained by differences in ionization dynamics. Numerical studies for*

materials such as fused silica and MgF_2 [44, 45, 47] showed a similar asymmetry, as a consequence of multi-photon absorption being more efficient at shorter wavelengths—even for the same multi-photon order, as shown in Si [48, 49]—whereas avalanche ionization becomes more efficient at longer wavelengths. The latter trend is directly described by the Drude model, which predicts that the inverse Bremsstrahlung heating rate scales as λ^2 , so free carriers gain energy more efficiently from longer-wavelength fields. In the down-chirped configuration, the blue spectral components (shorter wavelengths) arrive earlier in the pulse, producing a higher initial free-carrier density that enhances avalanche ionization by the subsequent red components (longer wavelengths). As a secondary mechanism, plasma-induced anomalous dispersion may further strengthen this effect by causing the blue components to catch up with the red ones, compressing the pulse and increasing its peak intensity.” (Primary Manuscript, page 5)

The new Section titled “S3.4 Chirped temporal profiles” has been added to the Supplementary Information. This Section includes the new Fig. S8 showing fine characterizations of the input laser spectrum as well as calculations of the temporal intensity profile in the up- and down-chirp configurations, compared to bandwidth-limited pulses.

Reviewer #1, Point 6

Reviewer #1: Regarding the explanation of Figure 4b, how to determine the factor of 2.4 in F_p between up- and down-chirped pulses at line 259?

Authors: The factor of 2.4 is obtained by taking the ratio between the peak fluence values determined for down-chirped pulses ($F_p^{\text{down}} = 1.17 \text{ J/cm}^2$) and up-chirped pulses ($F_p^{\text{up}} = 0.49 \text{ J/cm}^2$).

Revisions brought to the manuscript: The peak fluence values now appear in the corresponding figure [Fig. 5(b) in the revised manuscript]. The factor of 2.4 is highlighted in Fig. 5(b).

Reviewer #1, Point 7

Reviewer #1: What is the theoretical basis for the wavelength selection during the switching of multiphoton absorption orders at line 318?

Authors: The theoretical basis for the wavelength selection first relies on the band gap E_g of the considered materials (InP: $E_g = 1.34 \text{ eV}$; GaAs: $E_g = 1.43 \text{ eV}$). We acknowledge and correct an error in the originally submitted manuscript, where the band gap of InP was incorrectly given as $E_g = 1.24 \text{ eV}$ instead of $E_g = 1.34 \text{ eV}$. Second, the photon energy is $\hbar\omega = 0.63 \text{ eV}$ at a wavelength of $\lambda = 1960 \text{ nm}$, and $\hbar\omega = 0.80 \text{ eV}$ at a wavelength of $\lambda = 1555 \text{ nm}$. The multi-photon absorption order N is given by

$$N = 1 + \lfloor E_g/\hbar\omega \rfloor, \quad (\text{R4})$$

where $\lfloor \cdot \rfloor$ denotes the floor function, \hbar is the reduced Planck constant, $\omega = 2\pi c/\lambda$ is the laser angular frequency, and c is the speed of light in vacuum. As a result, for $\lambda = 1960 \text{ nm}$, $N = 2$ for Si and Ge, and $N = 3$ for InP and GaAs. In Fig. 5(d)–(f) of the revised Primary Manuscript, $\lambda = 1555 \text{ nm}$ which results in $N = 2$ for InP and GaAs.

Revisions brought to the manuscript: The band gap value of InP has been changed to $E_g = 1.34 \text{ eV}$ in Fig. 1, as well as in Table S1 (Supplementary Information), where Ref. [S12] has been added.

Reviewer #1, Point 8

Reviewer #1: The paper proposes three methods of optimizing energy deposition including increasing pulse duration, using a down-chirped pulse and achieving a higher multiphoton absorption order. Is there a hierarchy or relative priority among these methods? Furthermore, are there any quantitative metrics that can be used to evaluate the degree of energy deposition optimization?

Authors: This is a very interesting point from Reviewer #1, and we are grateful for this comment. Using the peak fluence F_p as the optimization metric, the three methods can be ranked as follows:

1. **Increasing the multi-photon absorption order.** The most impactful strategy is to shift to longer excitation wavelengths, thereby increasing the MPA order. From the data in the revised manuscript, Fig. 5(e), using 3-photon absorption instead of 2-photon absorption increases F_p by a factor of 15.0 in InP and 17.6 in GaAs.
2. **Increasing the pulse duration.** Optimizing the pulse duration ranks second. From Fig. 3(a), using $\tau = 25$ ps instead of $\tau = 275$ fs increases F_p by factors of 8.7 (Si), 4.6 (Ge), 5.9 (InP), and 4.0 (GaAs).
3. **Using down-chirped pulses.** Chirp control ranks third. From Fig. 3(b), using down-chirped instead of up-chirped pulses increases F_p by a factor of 2.4 in Si for $\tau = 3$ ps.

This hierarchy is now explicitly highlighted in the conclusion. As a side note, if the final application concerns the formation of permanent modifications inside the material (e.g., ultrafast laser writing), the modification threshold or modification volume would be a more relevant optimization metric than F_p .

Revisions brought to the manuscript: The following statement has been added to the conclusion:

“Among the tested optimization strategies, increasing the multi-photon absorption order by using longer wavelengths proved most effective for enhancing the peak fluence, followed by increasing the pulse duration, and finally by using down-chirped pulses. This hierarchy, quantified in the present work, provides practical guidance for tailoring ultrafast laser parameters in semiconductors.” (Primary Manuscript, page 7)

In agreement with Reviewer #2, Point 2, a new section titled “S4.5 **Summary tables**” has been added to the Supplementary Information. This new section contains **Tables S7–S10** where the peak fluence F_p , the effective critical power P_{cr}^{eff} , and the multi-photon absorption coefficient β_N^{eff} are given for different wavelengths λ , multi-photon absorption orders N , pulse durations τ , and chirps. These tables which contain the results from the revised Primary Manuscript, Figs. 3 and 5, give readers a comprehensive comparison of the effects of different regimes.

Reviewer #1, Point 9

Reviewer #1: The introduction inadequately covers the applications of ultrafast laser filamentation in gases.

Authors: We agree with Reviewer #1 that a detailed list of applications is not necessary in the introduction. We have therefore shortened this part while keeping references to representative studies that demonstrate how the fundamental understanding of filamentation has enabled important applications.

Revisions brought to the manuscript: The original statement in the introduction:

“In gases, the remarkable properties of filaments have led to a plethora of applications including light detection and ranging (LIDAR) [3], spectroscopy [4], terahertz wave generation [5], fog clearing [6], air waveguides [7, 8], and lightning guidance [9].”

has been revised to:

“In gases, the remarkable properties of filaments have led to a plethora of applications [3–9]” (Primary Manuscript, page 1)

Reviewer #1, Point 10

Reviewer #1: In the Materials & methods, the section of propagation calculations mainly describes the calculation process of F_{max} , P_{cr}^{eff} , β_2^{eff} , β_3^{eff} , the title should be revised for clarity.

Authors: We agree with Reviewer #1 and have revised the title of the corresponding section to better reflect its content.

Revisions brought to the manuscript: In the Materials & methods, the original title of the section:

“Propagation calculations.”

has been revised to:

“Determination of the key nonlinear optical coefficients.” (Primary Manuscript, Materials & methods, page 8)

Reviewer #1, Point 11

Reviewer #1: Besides ultrafast laser writing, supercontinuum generation during ultrashort laser pulses propagation in transparent solid is also widely concerned. Could the authors conduct a discussion regarding the impact of energy deposition on the supercontinuum generation?

Authors: We are grateful to Reviewer #1 for this particularly relevant and inspiring point. The work presented in our manuscript is indeed not only limited to ultrafast laser writing, but is also valuable in various fields including supercontinuum generation (SCG). While this is not the primary goal of our study, we have performed additional experiments to explore SCG in Si. These consist of transverse spectral measurements for various input pulse energies E_{in} and pulse durations τ . As expected, higher E_{in} and/or lower τ result in greater spectral broadening. While most studies on SCG in Si were performed with sub-ps pulses, our measurements suggest that spectral broadening persists even at single-digit ps pulse durations.

Besides SCG, accurate determination of the nonlinear optical coefficients in semiconductors is also crucial in other fields like backside processing, microelectronics security, THz wave generation, and high-harmonic generation (HHG). In all these fields, an inaccurate estimation of $P_{\text{cr}}^{\text{eff}}$ and/or $\beta_{2,3}^{\text{eff}}$ potentially results in suboptimal outcomes.

Revisions brought to the manuscript: A new section titled *“S4.6 Applications beyond laser direct writing”* has been added in the Supplementary Information. In this new section, the impact of a misestimation of $P_{\text{cr}}^{\text{eff}}$ and/or $\beta_{2,3}^{\text{eff}}$ in the aforementioned fields is discussed with corresponding references.

In this new section, the additional experimental results we obtained on SCG are included in a new figure (Fig. S17).

The original statement in the conclusion:

“The whole set of results allows us to envision future optimizations to improve the degree of control of energy deposition in the bulk of semiconductors, with various potential applications including in-chip ultrafast laser functionalization.”

has been revised to:

“The determination of effective key nonlinear coefficients summarized in the Supplementary Information, Section S4.5, is essential for predicting optimal conditions for ultrafast laser writing in semiconductors. Beyond this, these coefficients also provide critical insight into other fields, enabling advances for instance in backside processing, microelectronics security, THz wave generation, high-harmonic generation, and supercontinuum generation (see Supplementary Information, Section S4.6).” (Primary Manuscript, page 7)

The original statement has been added to the abstract:

“Beyond laser writing, the accurate determination of effective nonlinear parameters reported here is essential for predicting and optimizing semiconductor backside processing, microelectronics security, as well as high-harmonics, supercontinuum, and terahertz wave generation.” (Primary Manuscript, abstract, page 1)

Reviewer #1, Point 12

Reviewer #1: Just like Figure 3b, please add the error bars and corresponding meanings for Figures 3a and 3c.

Authors: We thank Reviewer #1 for spotting this oversight.

Revisions brought to the manuscript: The error bars have been added to Fig. 3(a) and (c).

To clarify the meaning of the error bars in Fig. 3(b), the original statement:

“The critical pulse energy E_{cr} , which delimits the two propagation regimes is thus determined as the average between the highest input pulse energy for which the propagation is linear [$E_{\text{in}}^- = 3 \text{ nJ}$ in Fig. S9(b)], and the lowest input pulse energy for which the propagation is nonlinear [$E_{\text{in}}^+ = 6 \text{ nJ}$ in Fig. S9(b)].”

has been revised to:

“The critical pulse energy E_{cr} , which delimits the two propagation regimes is thus determined as the average between the highest input pulse energy for which the propagation is linear [$E_{\text{in}}^- = 3 \text{ nJ}$ in Fig. S13(b)], and the lowest input pulse energy for which the propagation is nonlinear [$E_{\text{in}}^+ = 6 \text{ nJ}$ in Fig. S13(b)]. The effective critical power shown in the Primary Manuscript, Fig. 3(b) is evaluated for different pulse durations τ as

$$P_{\text{cr}}^{\text{eff}} = 0.88 \frac{T_F E_{\text{cr}}}{\tau}, \quad (S15)$$

where T_F is the Fresnel transmission coefficient. The uncertainty in $P_{\text{cr}}^{\text{eff}}$ values is calculated as

$$\varepsilon = 0.88 \frac{T_F |E_{\text{in}}^+ - E_{\text{cr}}|}{\tau} = 0.88 \frac{T_F |E_{\text{in}}^- - E_{\text{cr}}|}{\tau}. \quad (S16) \quad \text{(Supplementary Information, page 8)}$$

To clarify the meaning of the error bars in Fig. 3(a), the following sentence has been added:

“Therefore, F_p values are reported with an uncertainty of $\pm 15\%$.” (Supplementary Information, page 4)

To clarify the meaning of the error bars in Fig. 3(c), the following sentence has been added:

“Accounting for the uncertainty in $P_{\text{cr}}^{\text{eff}}$, the error on β_2^{eff} and β_3^{eff} values is estimated at $\pm 50\%$.” (Supplementary Information, page 11)

Response to Reviewer #2

Reviewer #2, Foreword

Reviewer #2: In the manuscript “The universality of filamentation-caused challenges of ultrafast laser energy deposition in semiconductors”, the authors analyze the influence of material properties, wavelength, pulse duration, and chirp of ultrashort laser pulses on the process of laser energy deposition in semiconductors. The manuscript has undeniable practical significance from the perspective of integrated 3D silicon photonics, as well as considerable fundamental importance, since the interaction of ultrashort laser pulses with semiconductors remains still insufficiently studied. The work can be conditionally divided into several parts: the first investigates the influence of the material on the fluence profile within the volume of the semiconductor; the second discusses the impact of pulse duration and chirp on this distribution; and the last examines the influence of wavelength. The quality of the experimental and numerical modeling work is at a very high level. The literature review is sufficiently modern, and the results are novel. The work is potentially of interest to a broad readership. Individually, the sections mentioned above are well-organized, have clear conclusions, and apart from some presentation features, I have practically no comments about them. However, combined, they do not form a cohesive single work, sometimes presenting too much information and raising questions that are not fully answered.

Authors: We sincerely thank Reviewer #2 for the thorough reading of our manuscript and the constructive feedback. We are grateful for the positive assessment of our work, particularly regarding “*the quality of the experimental and numerical modeling work*” and the novelty of the results. The reviewer’s insightful comments have been invaluable in improving not only the clarity and presentation, but also the cohesiveness of the revised manuscript, ensuring that the individual sections now form a more unified narrative. We address each concern in a point-by-point manner below. For clarity, reviewer comments are shown in **blue**, our responses in **green**, and modifications in the revised manuscript in **red**.

Reviewer #2, Point 1

Reviewer #2: Firstly, at the beginning of the manuscript, in Fig. 1, the authors provide information on the bandgap, nonlinear and linear refractive indices, from which it follows that Ge differs most strongly from the other three samples, InP and GaAs have similar properties, and Si properties are somewhat intermediate (except for the nonlinear refractive index). However, later in the text, the maximum fluence is achieved precisely in silicon, and overall, the dependencies are most pronounced there (properties are least pronounced in germanium).

Authors: Among all tested materials, Ge exhibits the smallest band gap, which is consistent with Fig. 3(b), where Ge shows the lowest effective critical power P_{cr}^{eff} . This means that a lower input pulse energy E_{in} and/or longer pulse duration τ is sufficient to trigger nonlinear effects in Ge. In terms of nonlinear absorption mechanism, Ge behaves similarly to Si, since two-photon absorption (2PA) is the dominant process in both. In Fig. 3(c), the effective two-photon absorption coefficient β_2^{eff} is higher in Ge than in Si, consistent with the smaller band gap of Ge. This indicates that lower intensities are required to generate a given free-carrier density in Ge compared to Si. However, the higher β_2^{eff} also leads to stronger prefocal absorption, which depletes the beam before the nonlinear focusing can fully develop, and limits the achievable peak fluence F_p compared to Si.

Reviewer #2, Point 2

Reviewer #2: Secondly, the article varies several parameters simultaneously: wavelength, energy, duration, chirp, and material. As a result, after multiple readings, it creates an impression of some excessive clutter, and it is unclear which particular effect dominates in each case.

Based on these two points, at present, the open question remains: which specific material properties influence the propagation and absorption of ultrashort laser pulses in the mid-IR range in semiconductors? It seems there is no monotonic dependence on the bandgap, linear, or nonlinear refractive indices, although from general principles, one might expect that a larger bandgap and lower refractive indices (both linear and nonlinear) would lead to increased fluence. However, the data presented by the authors do not demonstrate this. Therefore, it might be advisable for the authors to consider a more comprehensive comparison of the effects of different

regimes, perhaps in the form of one or several tables, or alternatively, splitting the work into multiple publications (even materials on silicon and/or germanium seem sufficient for a high-quality paper). As it stands, the number of samples rather confuses than adds value.

Authors: We have reorganized the manuscript to reduce the impression of excessive clutter.

The question raised by Reviewer #2 on which specific material properties influence ultrashort laser pulse propagation and absorption in the mid-IR is highly relevant. As noted, there is unfortunately no universal single-parameter rule, since nonlinear propagation results from the combined action of several physical effects. Nonlinear refraction and absorption depend on the band structure, while the ratio between the laser photon energy and the band gap determines the multi-photon absorption (MPA) order. The direct or indirect nature of the band gap also plays a role, as indirect transitions require phonon participation, making the MPA process less probable. Additional factors that were not varied in our study, such as dopant type, doping concentration, and crystal orientation, can also affect the effective nonlinear coefficients.

Despite these material-specific quantitative differences, a key finding of our work is that all tested semiconductors follow the same qualitative trends when E_{in} and τ are varied [see Figs. 2(d), 3, and 4 in the revised Primary Manuscript].

We are honored by the comment of Reviewer #2 that our results could be split into multiple high-quality papers. However, we believe that keeping all results together in a single publication provides a unique and comprehensive overview of ultrashort pulse filamentation in semiconductors, making it more valuable as a reference.

Finally, we thank Reviewer #2 for suggesting a summary in tabular form. We have added a dedicated section with summary tables to the Supplementary Information.

Revisions brought to the manuscript: To avoid the impression of some excessive clutter, **Fig. 3 in the originally submitted manuscript has been split into two figures (Figs. 3 and 4 in the revised Primary Manuscript).**

A new section titled “S4.5 **Summary tables**” has been added to the Supplementary Information. This new section contains **Tables S7–S10** where the peak fluence F_p , the effective critical power $P_{\text{cr}}^{\text{eff}}$, and the multi-photon absorption coefficient β_N^{eff} are given for different wavelengths λ , multi-photon absorption orders N , pulse durations τ , and chirps. These tables, which present the results from the revised Primary Manuscript, Figs. 3 and 5, allow readers to make a comprehensive comparison of the effects of different regimes, and to use these parameters for various applications, as suggested by Reviewer #2.

The original statement in the conclusion:

“The whole set of results allows us to envision future optimizations to improve the degree of control of energy deposition in the bulk of semiconductors, with various potential applications including in-chip ultrafast laser functionalization.”

has been revised to:

“The determination of effective key nonlinear coefficients summarized in the Supplementary Information, Section S4.5, is essential for predicting optimal conditions for ultrafast laser writing in semiconductors. Beyond this, these coefficients also provide critical insight into other fields, enabling advances for instance in backside processing, microelectronics security, THz wave generation, high-harmonic generation, and supercontinuum generation (see Supplementary Information, Section S4.6).” (Primary Manuscript, page 7)

Reviewer #2, Point 3

Reviewer #2: I believe that the information about the influence of chirp is somewhat excessive for the main text of a data-heavy publication. Perhaps it could be moved to supplementary materials?

Authors: We appreciate this suggestion from Reviewer #2. However, we note that Reviewer #3 explicitly requested both clarification and emphasis on chirp-related effects (Points 2 and 14). These results—shown in the revised Primary Manuscript, Figs. 5(a)–(c)—are critical to demonstrate that energy deposition is influenced not only by the central wavelength [Figs. 5(d)–(f)] but also by the temporal ordering of spectral components. For this reason, we believe retaining this material in the main text is essential to preserve the completeness of the discussion and address simultaneously the points of both reviewers.

Reviewer #2, Point 4

Reviewer #2: Fig. 3g,h are extremely poorly readable. Extracting information from them in their current form is very difficult.

Authors: We acknowledge that the original 3D plots were challenging to interpret. To improve clarity, the xy -integrated measurements are now shown in a separate figure in the Primary Manuscript, and two-dimensional plots extracted from the 3D data have been added to the Supplementary Information. The revised manuscript also retains Supplementary Videos 1 and 2, which provide rotating visualizations of the 3D plots.

Revisions brought to the manuscript: In the revised manuscript, Figs. 3(a)–(c) and 3(e)–(h) are presented as two separate figures (Figs. 3 and 4, respectively). Figure 3(d) from the original submission has been removed.

New figures which show 2D plots extracted from the 3D plots have been added to the Supplementary Information (Figs. S11 and S12). The following statement has been added to the Supplementary Information:

“The evolution of f_E and L_{abs} as a function of E_{in} and the corresponding fits according to Eq. (S13) are displayed as two-dimensional graphs in Figs. S11 and S12, respectively.” (Supplementary Information, page 8)

The following statement has been added to the caption of the new Fig. 4:

“Supplementary Videos 1 and 2 offer rotating visualization of the 3D plots in (c) and (d), respectively, and two-dimensional representations of these 3D plots are shown in Supplementary Information, Section S4.2.” (Primary Manuscript, Fig. 4 caption)

Reviewer #2, Point 5

Reviewer #2: Due to the large number of varying parameters and numerous figures, I recommend explicitly indicating the sample and the wavelength of the laser pulse on each figure. Working with the current presentation is quite challenging.

Authors: We thank Reviewer #2 for this helpful suggestion and fully agree that clearer figure labeling improves readability. In the revised Primary Manuscript and Supplementary Information, the material, laser wavelength, and pulse duration are now explicitly indicated on every figure where this information is relevant. This ensures that readers can immediately identify the experimental conditions without referring back to the text.

Revisions brought to the manuscript: When this was not explicit in the originally submitted manuscript, we have added **information about the tested material, wavelength and pulse duration**. This information is now indicated consistently across all figures in the revised manuscript.

Reviewer #2, Point 6

Reviewer #2: In Fig. 4d, the change in wavelength clearly affects the divergence angle of the “angel wings,” see the figure below (in the pdf version of the review). Interestingly, in Figs. 4a, 2b, these angles are approximately the same. Is this caused by the change in the refractive index at the interfaces when changing the wavelength and, consequently, the effective focusing?

Authors: Once again, this is a very interesting point from Reviewer #2, and we are thankful for this. After careful verification in all tested configurations, the angle of the “angel wings” follows the expectations from the linear propagation regime, where the half-angle of the cone of light reads $\theta = \arcsin(\text{NA}/n_0)$, where $\text{NA} = 0.40$ is the numerical aperture of the focusing objective lens, and n_0 is the linear refractive index of the medium. For InP and GaAs, the wavelength difference between 1555 nm and 1960 nm results in a refractive index change on the order of 3×10^{-2} . As a consequence, the difference in θ between the two wavelengths is less than 0.1° , which cannot explain the different angles of the angel wings.

In fact, the originally submitted manuscript contained a scale error in the images, and we apologize for this oversight. This has been corrected in Fig. 5(d) of the revised manuscript, where the angle of the “angel wings”

in GaAs is now consistent between the two wavelengths. This correction does not affect any measured values of F_p , P_{cr}^{eff} , $\beta_{2,3}^{eff}$.

Revisions brought to the manuscript: To better visualize that the angel wings follow the half-angle of the cone of light, θ appears in Fig. 2(b) of the revised manuscript.

The following statement has been added to the caption of the new Fig. 2:

“The vector k indicates the direction of propagation, and $\theta = \arcsin(NA/n_0)$ indicates the medium-dependent half-angle of the cone of light.” (Primary Manuscript, Fig. 2 caption)

The following statement has been added:

“The angel wings form an angle which follows the half-angle of the cone of light $\theta = \arcsin(NA/n_0)$.” (Primary Manuscript, page 2)

The images in Fig. 5(d) in the revised manuscript now show fluence distributions in GaAs to better visualize the conservation of the angle of the angel wings.

Reviewer #2, Point 7

Reviewer #2: In Fig. 2b, there is a clear trend of increasing defocusing effect when varying the sample from Si \rightarrow InP \rightarrow GaAs \rightarrow Ge. It might be worth changing the order to facilitate comparison of the impact regimes. Also, it seems that when changing the wavelength for the InP sample, the regime becomes similar to propagation in Ge. It would be advisable to align the energies presented in Figs. 2b, 4a, and 4c for better comparison.

Authors: We agree with Reviewer #2 on the trend of increasing defocusing when varying the sample from Si \rightarrow InP \rightarrow GaAs \rightarrow Ge. This order could thus indeed be considered in Fig. 2(b). However, Si and Ge are media where two-photon absorption (2PA) dominates at a wavelength of $\lambda = 1960$ nm, and InP and GaAs are media where three-photon absorption (3PA) dominates at the same wavelength. Moreover, Si and Ge exhibit an indirect band gap, while InP and GaAs exhibit a direct band gap. For these reasons, we prefer to keep these two subgroups of materials side by side, and we hope Reviewer #2 will understand our choice.

Reviewer #2 also rightly recalls that when the multi-photon absorption order is decreased from 3 to 2, the propagation in InP, and also in GaAs [see Fig. 5(d) of the revised manuscript and Reviewer #2, Point 6] becomes similar to Ge. This is due to the fact that prefocal absorption is much more pronounced for low multi-photon absorption orders, and this is actually one of the key messages of our study. Nevertheless, while 2PA dominates for Ge irradiated at $\lambda = 1960$ nm, and InP or GaAs irradiated at $\lambda = 1555$ nm, the comparison is not completely fair due to the indirect band gap of Ge and the direct band gaps of InP and GaAs. This implies that 2PA is less efficient in Ge than in InP and GaAs, which is confirmed in the summary Tables S7–S10 in the Supplementary Information (see Reviewer #2, Point 2). The enhanced absorption in InP and GaAs at $\lambda = 1555$ nm also implies that the employed input pulse energies E_{in} cannot be directly compared to Ge at $\lambda = 1960$ nm, as prefocal absorption governs the interaction at lower E_{in} values.

Revisions brought to the manuscript: The following statement has been added:

“Prefocal absorption is much more pronounced for 2PA than for 3PA. Due to their direct band gap, 2PA is even more efficient in InP and GaAs at $\lambda = 1555$ nm compared to Ge at $\lambda = 1960$ nm (see Fig. 2).” (Primary Manuscript, page 6)

Reviewer #2, Point 8

Reviewer #2: The authors use the term “filamentation,” but it seems that the length of the plasma channel is approximately the Rayleigh length, especially at low energies. Is it correct to use this term in this context?

Authors: This comment is absolutely correct, and it is in agreement with Reviewer #3, Points 1 and 8. We completely agree that a clear definition was missing in the originally submitted manuscript. This definition is now given at the beginning of the introduction.

Concerning the length of the plasma channel, let us first recall that nonlinear propagation imaging is time-integrated measurements, and we are thus unable to measure the plasma directly. However, we measure how light has been absorbed and refracted by the plasma. In the revised manuscript, we show in Fig. 4 (Primary Manuscript) and Fig. S12 (Supplementary Information) that the characteristic absorption length L_{abs} scales logarithmically with the intensity (through E_{in} and τ). Therefore, the length of the interaction zone increases with this parameter, but not spectacularly as in studies in air for example. This can be explained by our relatively tight focusing conditions, which imply that the Kerr effect not only competes with plasma defocusing, but also with diffraction. Nevertheless, resembling features were observed for looser focusing conditions. For instance, in Ref. [24], the numerical aperture NA is varied from 0.30 to 2.97, and in Ref. [37] filamentation was characterized for NA = 0.26

Revisions brought to the manuscript: The original statement in the introduction:

“Ultrafast laser filamentation is an extremely nonlinear propagation regime where nonlinear refraction competes with plasma effects [1, 2].”

has been revised to:

*“Ultrafast laser filamentation is an extremely nonlinear propagation regime **characterized by a dynamic balance between Kerr-induced self-focusing and plasma-induced defocusing [1, 2].**”* (Primary Manuscript, page 1)

Reviewer #2, Conclusion

Reviewer #2: In conclusion, I believe that this article undoubtedly deserves publication in Nature Communications, but only after major revision.

Authors: We thank Reviewer #2 once again for all their comments, which enabled us to drastically improve our manuscript, and we hope that the revisions we have made will be deemed satisfactory.

Response to Reviewer #3

Reviewer #3, Foreword

Reviewer #3: The paper by Chambonneau et al. is aiming to provide a broad vision about physical origins of limitation conditions in semiconductors when using ultrafast laser pulses, and therefore explain the inability, in many cases, to induce modification inside such materials. It treats an important scientific and engineering problem for the development of 3D laser writing technologies applicable to semiconductors.

The manuscript is generally pleasant for reading, contains original data and is well-structured. The authors present interesting discussions based on observations of nonlinear propagation effects across variety of semiconductors. However, from both technical and fundamental viewpoints (explained below), the paper does not succeed to convincingly demonstrate truly “universal” filamentation process which would be responsible for the observed limitations. Although strong nonlinearities are indeed expected in the studied regimes, authors’ choice to associate all observed effects under the same term of “filamentation” is, in my opinion, often excessive and leads to confusion.

Authors: We sincerely thank Reviewer #3 for the thorough reading of our manuscript and for the constructive feedback. We are grateful for the compliments noting that our “*manuscript is generally pleasant for reading, contains original data and is well-structured*”. We agree that the previous title may have been misleading, and we have revised it so that it now stresses the novelty aspects on the extracted effective nonlinear coefficients (see Reviewer #3, Point 2). The insightful comments raised by the reviewer have significantly contributed to enhancing the clarity, cohesiveness, and overall quality of the revised manuscript. We address each of the concerns point by point below. For clarity, reviewer comments are shown in **blue**, our responses in **green**, and modifications in the revised manuscript in **red**.

Reviewer #3, Point 1

Reviewer #3: Even if this terminology might be justified depending on a chosen definition (which is not clearly given in manuscript, see after), the discussions do not systematically analyze the problem in view of competing effects trapping the light propagation as traditionally expected for filamentation phenomena (e.g. Kerr-based and plasma nonlinearities in the simplest case). Filamentation in usual sense implies intensity clamping as result. But intensity clamping itself can occur in absence of filamentation, and can originate from different nonlinear mechanisms (plasma mirror effect as an example in a different context).

Authors: We agree with Reviewer #3 that, in general, there are always physical limits to energy deposition in a material using ultrashort laser pulses. Reviewer #3 rightly recalls that for surfaces, the plasma mirror effect is mainly responsible for these limitations. In the bulk of Si, this effect cannot occur because the laser-produced plasma is underdense [R1]. The clamping effect is due to the evolution of the interaction in the material within the prefocal region. This was observed in dielectrics under comparable conditions to those in our study [R2].

Revisions brought to the manuscript: To clearly define filamentation, the original statement in the introduction:

“Ultrafast laser filamentation is an extremely nonlinear propagation regime where nonlinear refraction competes with plasma effects [1, 2].”

has been revised to:

*“Ultrafast laser filamentation is an extremely nonlinear propagation regime **characterized by a dynamic balance between Kerr-induced self-focusing and plasma-induced defocusing [1, 2].**”* (Primary Manuscript, page 1)

Reviewer #3, Point 2

Reviewer #3: Furthermore, paper does not bring new technique — it reproduces previous measurements done in silicon for other semiconductors. Also, no new physical concept or mechanism is directly demonstrated. Although the dependence on parameters like the chirp of the pulses is shown and is indeed interesting, authors themselves admit they do not provide a concrete solution or a clear new path to overcome the issue. They write that “the whole data set allows [them] to envision future optimizations”, but such statement stays vague and

does not bring truly novel idea comparing to earlier reports.

Authors: We agree with Reviewer #3 that the technique we employ is not new, and we never claimed this. In fact, in the originally submitted manuscript, we explained that nonlinear propagation imaging was initially developed for water, adapted to dielectrics, and then applied to examine nonlinear propagation of light in Si. The essential aspect of our study is the application of this simple and elegant technique to determine realistic nonlinear optical coefficients, which has never been done before. As noted by Reviewer #3, we did not restrict our study to Si, but performed experiments in different semiconductors.

We respectfully disagree with the following statement: “no new physical concept or mechanism is directly demonstrated”. One of the key original findings of our study is that the nonlinear optical coefficients are orders of magnitude higher than expected from literature values obtained with the z-scan method. Moreover, we establish the temporal scaling laws for these nonlinear coefficients, and we demonstrate that longer pulse durations lead to higher nonlinear refraction and absorption—a phenomenon never studied before in semiconductors. The temporal dependence of the effective critical power is explained by delayed Kerr nonlinearity, previously uncharted for semiconductors. The model we devise which is based on a modified Marburger formula and allows us to extract the multi-photon absorption coefficients of different orders, and simultaneously predict the nonlinear focal shift, is novel. Finally, we propose three ways to increase the peak fluence F_p in semiconductors, which consist of employing (1) longer pulses, (2) down-chirped pulses, and (3) higher multi-photon absorption orders.

Finally, we agree with Reviewer #3 on the vague statement, and we have revised it accordingly. In the revised manuscript, applications beyond ultrafast laser writing are discussed, in line with Reviewer #1, Point 11

Revisions brought to the manuscript: To put the emphasis on the extreme nonlinear coefficients determined in our study, the originally submitted title: “*The universality of filamentation-caused challenges of ultrafast laser energy deposition in semiconductors*”

has been revised to:

“*Extreme optical nonlinearities unveiled by ultrafast laser filamentation in semiconductors*” (Primary Manuscript, page 1)

Reviewer #3, Point 3

Reviewer #3: Especially, I find not well addressed the question on high-order multiphoton absorption in comparison to a recent study on silicon using mid-IR femtosecond pulses (see ref.31). This previous study should be more critically discussed, because it suggests possibility for in-volume modification in silicon and may contradict the claim of an “universal” limitation due to filamentation. The current paper can be seen as directly supporting this alternative route, or contrarily as not fully compatible with the ‘universal’ picture stressed in it, making the delivered message unclear on this aspect.

Authors: The study in Ref. [31] reports internal modifications in Si with 160-fs pulses at 4600 nm, corresponding to the 5-photon absorption regime. To the best of our knowledge, these are the first modifications produced solely in the bulk of Si through a plane surface with such short pulses. This result does not contradict our conclusions; rather, it supports our finding in the revised manuscript [Fig. 5(d)–(f)] that higher multi-photon absorption (MPA) orders can enhance energy deposition. In the mid-IR, the longer wavelength increases the MPA order, which delays plasma defocusing and allows the fluence modification threshold to be reached even in the filamentation regime. Moreover, while mid-IR pulses lead to lower electron densities, the deposited energy is higher because of the enhanced efficiency of inverse Bremsstrahlung absorption (IBA) [R3]. We thank Reviewer #3 for the suggestion to improve the discussion of our results compared to Ref. [31], and have incorporated this comparison into the revised manuscript.

More generally, there is no incompatibility between filamentation and bulk modifications. As exemplified in Ref. [R2] for fused silica, modifications can occur in the filamentation regime whenever the local fluence exceeds the modification threshold. The specificity of semiconductors lies in their narrow band gap, which increases prefocal absorption at shorter wavelengths, making it far more difficult to reach the modification threshold compared to wide-gap dielectrics.

Revisions brought to the manuscript: The original statement:

“However, an important result is that the peak fluence F_p is one order of magnitude higher in the 3PA regime compared to the 2PA regime (maximum ratio of 13 and 21 for InP and GaAs, respectively), suggesting that

higher multi-photon absorption orders are beneficial to exalt the interaction.”

has been revised to:

“However, an important result is that the peak fluence F_p is more than one order of magnitude higher in the 3PA regime compared to the 2PA regime, suggesting that higher multi-photon absorption orders are beneficial to tailor energy deposition. This is consistent with results obtained in Si with 160-fs pulses in the 5-photon absorption regime [31], where permanent bulk modifications were achieved due to reduced prefocal absorption and the ability to reach the modification threshold. Although mid-infrared pulses generate lower electron densities, the deposited energy is increased through more efficient inverse Bremsstrahlung absorption [16]. These findings indicate that high-order multi-photon absorption can, under specific conditions, relax filamentation-induced constraints, whereas at shorter wavelengths filamentation still limits energy deposition in semiconductors.” (Primary Manuscript, page 6)

Reviewer #3, Point 4

Reviewer #3: My criticisms should not be understood as a questioning on the merit and interest of the work. The presented data are valuable and of interest for a specialized community. However, in my view, the paper does not offer sufficiently significant advancement in terms of new mechanism, concept, or technique to make it appropriate for a broad audience journal as Nature Communications.

Authors: We emphasized in the originally submitted manuscript that the main application of our study was ultrafast laser writing. Because of this focus, we understand that Reviewer #3 believes our manuscript is only valuable for a “specialized community” and thus not appropriate for a broad audience journal such as Nature Communications. In fact, the determination of effective nonlinear coefficients is applicable in numerous fields beyond ultrafast laser writing. These fields include for instance backside processing, microelectronics security, THz wave generation, high-harmonic generation, and supercontinuum generation, which are now detailed in the revised manuscript.

Revisions brought to the manuscript: A new section titled “S4.6 Applications beyond laser direct writing” has been added in the Supplementary Information. In this section, the impact of a misestimation of P_{cr}^{eff} and/or $\beta_{2,3}^{eff}$ is discussed for backside processing, microelectronics security, THz wave generation, high-harmonic generation, and supercontinuum generation, with corresponding references.

In this new section, the additional experimental results we obtained on supercontinuum generation are included in a new figure (Fig. S17).

The original statement in the conclusion:

“The whole set of results allows us to envision future optimizations to improve the degree of control of energy deposition in the bulk of semiconductors, with various potential applications including in-chip ultrafast laser functionalization.”

has been revised to:

“The determination of effective key nonlinear coefficients summarized in the Supplementary Information, Section S4.5, is essential for predicting optimal conditions for ultrafast laser writing in semiconductors. Beyond this, these coefficients also provide critical insight into other fields, enabling advances for instance in backside processing, microelectronics security, THz wave generation, high-harmonic generation, and supercontinuum generation (see Supplementary Information, Section S4.6).” (Primary Manuscript, page 7)

The following statement has been added to the abstract:

“Beyond laser writing, the accurate determination of effective nonlinear parameters reported here is essential for predicting and optimizing semiconductor backside processing, microelectronics security, as well as high-harmonics, supercontinuum, and terahertz wave generation.” (Primary Manuscript, abstract, page 1)

Reviewer #3, Point 5

Reviewer #3: The general claim about broad potential impact of the findings is not clearly supported. The suggested advances in understanding and controlling nonlinear propagation regimes in semiconductors remain vague. Authors refer to applications in sensing, quantum engineering, health care, etc., but without concrete

connection or perspective to actual implementations.

Authors: Here, Reviewer #3 refers to the following original statement in the abstract:

“Light propagation in semiconductors is the cornerstone of emerging disruptive technologies holding considerable potential to revolutionize telecommunications, sensors, quantum engineering, healthcare, and artificial intelligence. Sky-high optical nonlinearities make these materials ideal platforms for photonic integrated circuits.”

As a matter of fact, the applications in telecommunications, sensors, quantum engineering, healthcare, and artificial intelligence are all directly linked with semiconductor-based photonic integrated circuits (PICs):

- **Telecommunications:** Silicon photonics and InP-based PICs are the backbone of high-speed optical transceivers and data center interconnects.
- **Sensing:** Silicon nitride (SiN) and InP PICs are used in LiDAR, spectroscopic sensors, and integrated biosensing platforms.
- **Quantum engineering:** GaAs, InP, and Si-based PICs support integrated quantum light sources, waveguides, and detectors.
- **Healthcare:** Semiconductor PIC biosensors are used for point-of-care diagnostics, OCT imaging, and lab-on-chip devices.
- **Artificial intelligence:** Silicon photonics is being developed for optical neural networks and photonic tensor processors.

The findings in our manuscript allow us to envision selective internal control of properties of semiconductors.

Revisions brought to the manuscript: To clarify this aspect, the following statement in the abstract:

“Light propagation in semiconductors is the cornerstone of emerging disruptive technologies holding considerable potential to revolutionize telecommunications, sensors, quantum engineering, healthcare, and artificial intelligence. Sky-high optical nonlinearities make these materials ideal platforms for photonic integrated circuits.”

has been revised to:

*“Light propagation in semiconductors is the cornerstone of emerging disruptive technologies, **such as photonic integrated circuits with the potential to** revolutionize telecommunications, **sensing**, quantum engineering, healthcare, and artificial intelligence. Sky-high optical nonlinearities make these materials ideal platforms for **compact, multifunctional photonic devices.**”* (Primary Manuscript, abstract)

The original statement:

“Two indirect (Si and Ge) and two direct (InP and GaAs) band-gap semiconductors have been selected (see the inset in Fig. 1). Because of their narrow band gaps ($E_g < 1.5$ eV), these media exhibit high linear and nonlinear refraction ($n_0 > 3$, and $n_2 > 10^{-18}$ m²/W, respectively).”

has been revised to:

*“Two indirect (Si and Ge) and two direct (InP and GaAs) band-gap semiconductors have been selected (see the inset in Fig. 1). **Besides technological importance due to their widespread use in microelectronics, photovoltaics, sensing, and quantum engineering, these materials exhibit cubic crystal structures, a property that minimizes anisotropy effects during nonlinear propagation.** Because of their narrow band gaps ($E_g < 1.5$ eV), these media exhibit high linear and nonlinear refraction ($n_0 > 3$ and $n_2 > 10^{-18}$ m²/W, respectively).”* (Primary Manuscript, page 2)

Reviewer #3, Point 6

Reviewer #3: I understand the motivation is to enable future modification inside semiconductors using ultrashort laser pulses. However, as stated in the introduction, such modifications have already been demonstrated in particular with longer and/or mid-IR pulses. Also, remarkable results exist using nanosecond pulses [refs 28, 29]. The paper does not explain sufficiently what specific new regime is targeted, and how it differs from these previous realizations. It does not describe what is exactly required to reach applications (e.g. specific material transformation) and how the current results are making critical step forward.

Authors: We must emphasize that our motivation is not limited to producing modifications inside semiconductors using ultrashort laser pulses. Our primary objective is to gain a deeper understanding of nonlinear

propagation of ultrashort laser pulses inside semiconductors. To quantify this, we establish effective key non-linear parameters and their temporal scaling laws, and we propose methods to increase the peak fluence in semiconductors.

Focusing specifically on laser writing, Reviewer #3 mentions Refs. [28, 29] where nanosecond laser pulses were used to functionalize silicon internally. These results are indeed interesting, which is why we cited them in the originally submitted manuscript. In Ref. [28], such pulses were used to inscribe various microstructures inside silicon, including optical elements and microfluidic channels. In Ref. [29], spatial light modulation was additionally employed to inscribe nanostructures elongated along the optical axis.

By contrast, our study employs ultrashort pulses. Compared to the nanosecond regime, the ultrashort pulse regime offers advantages such as limited heat-affected zones and reduced strain fields surrounding the modification [R4, R5]. This is the main motivation for the inscription of quantum circuits in glass with ultrashort laser pulses [R6]. Nevertheless, the large strain fields produced with nanosecond pulses can be advantageously exploited for waveguide inscription [R7] and for writing waveplates inside silicon, as demonstrated in Ref. [R8] by the same group as Refs. [28, 29].

Moreover, using Gaussian-shaped ultrashort laser pulses for structuring silicon internally would allow the inscription of nanostructures that are much more confined than those produced with Bessel beams in Ref. [29]. Finally, a major difference between the references mentioned by Reviewer #3 and our study is that our manuscript investigates four semiconductors, whereas Refs. [28, 29] focus solely on silicon.

Reviewer #3, Point 7

Reviewer #3: Authors claim they identify filamentation regimes common to all semiconductors, beyond Si. But some statements are not accurate: “However, in narrow-gap materials such as semiconductors, the understanding of ultrafast filamentation is to date limited to Si.” This is misleading — literature reports filamentation also in SiC, GaAs, etc. (e.g. <https://doi.org/10.1016/j.ceramint.2024.02.133>, <https://doi.org/10.1007/BF00897938>). Possibly, authors mean understanding of filamentation in regard to the limitations for ultrafast laser 3D machining applications and add that “...nothing indicates that the conclusions on filamentation in Si hold in other semiconductors. “ But even this statement appears questionable. After some literature check, I found no consensus that such limitations in Si are always due to filamentation. Intensity clamping is frequently observed, but is not necessarily result of filamentation process. Actually, as for dielectric studies the filamentation is rarely invoked in these works conducted with high NA focusing conditions. Also, somehow confirmed by Fig1, there is nothing which indicates a drastically different nonlinear propagation responses in other semiconductors than Si. The report is apparently a confirmation of this relatively intuitive aspect.

Authors: Here, Reviewer #3 cites two studies that supposedly address filamentation in SiC [R9], and GaAs [R10]. First, in the study in SiC [R9], filamentation occurs *in air*, and not inside the semiconductor. These air filaments are then used to ablate SiC, as highlighted in this article: “*The laser was focused with a stationary lens ($f = 30\text{ cm}$) to generate filament in air which was then used to groove SiC CMC.*” Therefore, this paper is not relevant for our study on filamentation in semiconductors. Second, in the article in GaAs [R10], filamentation is indeed studied inside the semiconductor; however, this is a purely theoretical study. Moreover, this work was published in 1980—five years before the advent of chirped pulse amplification in 1985 [R11] which has made femtosecond lasers broadly accessible for laboratory use. Consequently, no specific pulse duration was given in this article. Therefore, this paper is also not relevant for our study.

According to Reviewer #3, “*for dielectric studies the filamentation is rarely invoked in these works conducted with high NA focusing conditions*”. We respectfully disagree with this. Many groups working on laser-produced modifications in dielectrics with high numerical aperture use the “filamentation” terminology. For instance, in the case of fused silica, Ref. [R12] studied filamentation with $NA = 0.05\text{--}0.25$, Ref. [R2] studied filamentation with $NA = 0.50$, and Ref. [R13] studied filamentation with $NA = 0.55$. In all these works, the Kerr effect competes not only with plasma defocusing but also with the natural divergence of the beam due to diffraction. As this divergence is a linear process which occurs for all pulse energies and durations, there is no contradiction with the definition of “filamentation” given in the revised manuscript, i.e., the “*dynamic balance between Kerr-induced self-focusing and plasma-induced defocusing*” (Primary Manuscript, page 1). As the conditions used in our study are close to the ones in Refs. [R2, R12, R13], we maintain the term “filamentation” throughout the manuscript.

Finally, Reviewer #3 mentions that, according to Fig. 1, “*there is nothing which indicates a drastically different nonlinear propagation responses in other semiconductors than Si*”, and that our study would be “*a confirmation of this relatively intuitive aspect*”. In Fig. 1, the nonlinear refractive index n_2 is given for different materials, including semiconductors. These n_2 values were mostly measured with standard techniques like z-scan. It is true that the order of magnitude for n_2 is similar for Si, Ge, InP and GaAs. Therefore, nonlinear propagation can be expected qualitatively at similar input intensities. However, by definition, filamentation is intrinsically linked with plasma formation through ionization. For the wavelength of $\lambda = 1960$ nm mainly used in our study, different ionization dynamics are expected between Si and Ge (2-photon absorption, indirect semiconductors), and InP and GaAs (3-photon absorption, direct semiconductors). One of the striking features of our study is that the effective critical power for nonlinearities decreases with the pulse duration τ , as a consequence of delayed Kerr nonlinearity. Moreover, the 2- and 3-photon absorption coefficients scale linearly with τ . Despite these similar trends, the quantitative differences are noteworthy, which all arise from the various intrinsic properties of the tested semiconductors. The determination of temporal scaling laws for effective nonlinear coefficients is paramount for a wide variety of applications beyond internal structuring (see Reviewer #3, Point 4).

Reviewer #3, Point 8

Reviewer #3: The paper lacks a clear definition of “filamentation”. The link between observed intensity clamping and filamentation is often suggested but the direct relationship remains unclear (see previous comments). Statements like “prefocal absorption hinders localized energy deposition” or “prefocal absorption governs the interaction” appear contradictory with the concept of a situation fully governed by filamentation.

Authors: As already mentioned in our response to Reviewer #3, Point 1, a clear definition of filamentation is now provided in the revised Primary Manuscript as the first sentence in the introduction.

We agree with Reviewer #3 that, for filamentation in transparent media, prefocal absorption is generally negligible. In semiconductors, however, this effect becomes significant due to low-order multi-photon absorption. Importantly, there is no contradiction between the occurrence of filamentation and the presence of prefocal absorption, which is an inevitable consequence of the high intensities reached during Kerr self-focusing in narrow-gap materials. The beam in semiconductors still exhibits strong spatial confinement, intensity clamping, and a Kerr–plasma balance. Prefocal losses simply shorten the filament length and reduce the energy available for localized deposition. This coexistence between filamentation and prefocal absorption is now explicitly stated in the introduction of the revised manuscript.

Revisions brought to the manuscript: In the Primary Manuscript, the original statement:

“In contrast with other media, nonlinear propagation effects in Si are disastrous when aiming for internal structuring, as the energy deposition is delocalized and saturates below the modification threshold due to intensity clamping [14–20].”

has been revised to:

“In contrast with other media, nonlinear propagation effects in Si are disastrous when aiming for internal structuring. Strong low-order multi-photon absorption before the geometrical focus (i.e., prefocal absorption) coexists with filamentation and leads to delocalized energy deposition, which saturates below the modification threshold due to intensity clamping [14–20].” (Primary Manuscript, page 1)

Reviewer #3, Point 9

Reviewer #3: Nonlinear regime categorization (Fig.2)

The proposed categorization of nonlinear propagation effects depending on laser power is interesting and represent the key contribution. However, the range of tested conditions seems not large enough to justify this claim of “universality”. The tested materials are relatively diverse, but most laser parameters are not varied significantly except pulse duration. In particular, it is unclear how robust the observations are with respect to different focusing conditions and wavelengths. For example, in dielectrics, high-NA focusing usually hinders self-focusing and filamentation. In addition, some features observed (like “angel” or “pearl necklace”) could be influenced by the apparent Fraunhofer diffraction patterns in the focused beam, especially when beam largely overfills the focusing optics. These patterns have side lobes, secondary on axis maxima and structures that may play major role in studied cases. These aspects are not discussed, and robustness to an ideal Gaussian focusing

for different NA is not demonstrated.

Authors: According to Reviewer #3, “most laser parameters are not varied significantly except pulse duration”. We respectfully disagree with this assessment, which also contradicts Reviewer #2, Point 2. In our study, **four materials** were tested (Si, Ge, InP and GaAs); the **pulse duration** was varied over nearly **two orders of magnitude** (275 fs – 25 ps); the **input pulse energy** spanned nearly **six orders of magnitude** (3 pJ – 1 μ J); **two wavelengths** (1960 nm and 1555 nm) were employed, corresponding to different multi-photon absorption orders for InP and GaAs; **two chirp configurations** (up and down) were used; and, for each condition, **2000 images** were recorded with 100-nm on-axis steps on the focusing lens.

Regarding the robustness of our observations with respect to different focusing conditions, this was extensively studied in 2017 in Ref. [24], where the numerical aperture was varied from NA = 0.30 up to NA = 2.97. The observed trends were identical, and intensity clamping occurred for all NA values. This study also demonstrated that the peak fluence increases linearly with NA. Given the extensive NA range already covered, we do not believe that additional experiments would provide new insights. For this reason, we fixed the NA to 0.40 in the present work. As shown in the Supplementary Information, Fig. S7, the choice for this NA was motivated by achieving the highest peak fluence without severe spherical aberration.

The point on Fraunhofer diffraction potentially influencing the observed features is very interesting, and we thank Reviewer #3 for raising it. In practice, when bell-shaped (e.g., Gaussian) beams are used, there is a trade-off between the desired effective NA and the presence of Fraunhofer diffraction patterns. If the beam overfills the entrance pupil of the lens, the peripheral portion is clipped, generating diffraction patterns; however, all angular components are still produced after the lens, and the NA corresponds to the manufacturer specifications. If the beam underfills the pupil, propagation is free from diffraction patterns, but the effective NA is smaller. The former approach—overfilling the aperture—is generally preferred in both research and industrial contexts to maximize peak fluence, and it is the one adopted in our experiments to obtain realistic and meaningful results. We agree that Fraunhofer diffraction can influence the fluence distribution morphology. However, as shown in Supplementary Information, Fig. S13(a), the positions of the secondary on-axis lobes depend on the input pulse energy, indicating that features influenced by diffraction evolve nonlinearly.

Revisions brought to the manuscript: In the Primary Manuscript, the original statement:

“*This pearl necklace morphology highlights the complex focusing and defocusing dynamics of filamentation.*” has been revised to:

“*This pearl necklace morphology highlights the complex focusing and defocusing dynamics of filamentation. In addition, morphologies for high E_{in} could be influenced by Fraunhofer diffraction patterns caused by the overfilling of the focusing optics. Noteworthy, the position of the secondary on-axis lobes shifts nonlinearly with the input intensity (see Supplementary Information, Section S4.3).*” (Primary Manuscript, page 2)

Reviewer #3, Point 10

Reviewer #3: Authors provide discussions based on pulse duration dependency. The scaling of saturation fluence $\propto \sqrt{\tau}$ is interesting, but used to claim that filamentation dominates even for long pulses (25 ps), which is unclear.

Authors: We agree with Reviewer #3 that further clarification is warranted. In fact, two features support the conclusion that filamentation still dominates even for long pulses (25 ps). First, the saturation of the maximum fluence F_{max} to a peak fluence value F_p is a hallmark of the filamentation regime. Second, the dependence of a fluence (F_p) on the pulse duration indicates that the propagation is nonlinear. In the absence of nonlinearities, the maximum fluence would be constant with respect to pulse duration, following $2E_{in}/(\pi w_0^2)$.

Revisions brought to the manuscript: The original statement:

“*Interestingly, in the tested pulse duration range ($\tau = 275$ fs – 25 ps), the peak fluence F_p scales as $\sqrt{\tau}$, which implies that increasing τ by a factor 100 results in an increase of F_p by only one order of magnitude. This highlights that, even for the longest pulses employed ($\tau = 25$ ps), filamentation still dominates the interaction.*” has been revised to:

“*Both the fluence saturation and the temporal dependence of F_p support the conclusion that filamentation still dominates the interaction even for the longest pulses employed ($\tau = 25$ ps). Interestingly, in the tested pulse duration range ($\tau = 275$ fs – 25 ps), the peak fluence F_p scales as $\sqrt{\tau}$.*” (Primary Manuscript, page 3)

Reviewer #3, Point 11

Reviewer #3: Determination of effective nonlinear parameters (critical power, multiphoton coefficients, nonlinear refractive index) is valuable, but remains very indirect. In such complex beam propagation (see above comments on beam structures), rigorous 3D simulations accounting for nonlinear effects would be necessary to confirm the extracted values are meaningful. This would be important to confirm the effective parameters are not simply compensating other effects (not accounted) due to the simplifications of the treatments to make valid the coefficient determinations for the studied conditions. This makes very difficult the comparisons with well-known values measured rigorously for well-defined conditions with specific methodologies.

Authors: Concerning the allegedly “*very indirect*” character of the effective nonlinear parameters, we respectfully disagree with Reviewer #3. The effective critical power for nonlinearities is determined as the condition at which a measurable deviation from the linear regime occurs. This is rather direct compared to the standard z-scan method (see for instance Ref. [R14]) which consists of (1) a predetermination of the multi-photon absorption coefficient β_N , (2) power-transmission measurements at multiple on-axis positions of the focusing lens, (3) data fitting with a theoretical model to extract the nonlinear refractive index n_2 , and from it deduce P_{cr} . Similarly, the effective multi-photon absorption coefficient β_N^{eff} is simply retrieved from the dependence of the nonlinear focal shift on the input pulse energy—a method no less direct than z-scan. Even if one assume that z-scan is a more direct technique, it leads to unrealistic $P_{\text{cr}}^{\text{eff}}$ and β_N^{eff} , in turn leading to a misestimation of the onset of nonlinear propagation, as highlighted in the Primary Manuscript, Figs. 3(b) and 3(c).

Reviewer #3 also highlights the absence of “*rigorous 3D simulations accounting for nonlinear effects*”. We agree that such simulations can be powerful for predicting fluence distributions in semiconductors. They can be performed with models based on the nonlinear Schrödinger equation (NLSE) [R3] or the unidirectional pulse propagation equation (UPPE) [R15, R16]. However, in prior works, very short pulse durations were used ($\tau = 250$ fs in Ref. [R3], and $\tau = 60$ fs in Refs. [R15, R16]). This ensures that the material response does not influence propagation or ionization—in other words, the electronic and lattice subsystems can be treated separately.

In contrast, in our study, the pulse duration is varied up to 25 ps, meaning the material response occurs during the pulse itself, since τ exceeds the typical electron–phonon scattering time (≈ 1 ps [R5]). Under these conditions, NLSE- and UPPE-based models cannot realistically simulate the interaction, especially for the longest picosecond pulses. This mismatch between theory and experiment has already been observed in Ref. [R17] for $\tau = 600$ –850-fs. Furthermore, recent results have shown that the electron–phonon scattering time in silicon is carrier-density dependent, increasing from 350 fs to 1.15 ps for fluences in the 0.6–1.5 mJ/cm² range [R18]. This makes modeling even more challenging in our case, where strong carrier-density gradients exist.

A truly rigorous 3D model that accounts for light propagation, ionization, carrier–phonon scattering, carrier mobility, thermal effects, band-gap shrinkage, and hydrodynamics does not currently exist. Developing such a model would demand enormous computational resources and is far beyond the scope of the present manuscript. Moreover, both NLSE and UPPE rely on a classical approach in which the band gap E_g is treated as constant. It is reasonable to expect that these approaches would not fully capture in-volume ultrafast laser–semiconductor interactions, and that quantum-level modeling (e.g., density functional theory [R19–R21]) could prove more accurate. To date, there is no successful implementation of such quantum-level models in nonlinear propagation codes in semiconductors. Thus, by providing experimentally determined effective nonlinear parameters, our work offers essential empirical constraints for such future models and thus makes a critical step toward a comprehensive theoretical description.

Revisions brought to the manuscript: The original statement in the abstract:

“*While nonlinear propagation of high-intensity ultrashort laser pulses has been extensively investigated in Si, other semiconductors remain uncharted.*”

has been revised to:

“*While nonlinear propagation of high-intensity ultrashort laser pulses has been extensively investigated in Si, projections for other semiconductors are impossible, given also the limitations of nonlinear propagation models to account correctly for the electron dynamics in such small band-gap materials.*” (Primary Manuscript, abstract, page 1)

Revisions brought to the manuscript: The original statement:

“*Nevertheless, nothing indicates that the conclusions on filamentation in Si hold in other semiconductors.*”

has been revised to:

“Nevertheless, the strong nonlinear absorption and the complex temporal electron dynamics observed during filamentation in Si together with the concomitant limitations of nonlinear propagation models to account correctly for the electron dynamics prohibit any generalization for other semiconductors.” (Primary Manuscript, page 1)

Reviewer #3, Point 12

Reviewer #3: Also, the methodology relies mostly on determinations of conditions for which deviations from linear propagation can be observed, but it is not strictly the critical power definition — it may for example detect early onset of detectable absorption, which is governed by multiple parameters not only the beam power. The critical power dependence to the pulse duration is well supported by theoretical considerations in supplementary information about a nonlinear refractive index with a delayed response. While it is interesting and it sounds, without comparison to rigorous and complete propagation simulations I consider the authors should avoid to report on measurements on nonlinear coefficients in the studied conditions and try to compare to established values for different conditions. The employed methodology is highly performing for direct monitoring of intense beam behaviors and provides interesting observations. However, I do not consider it is the most appropriate methodology to determine nonlinear coefficients with precision.

Authors: We agree with Reviewer #3 that the effective critical power $P_{\text{cr}}^{\text{eff}}$ determined with our method is not strictly the critical power *for self-focusing*, but, more generally, the critical power *for nonlinearities*. This is precisely why, in the originally submitted manuscript, the corresponding section in the Supplementary Information was titled *“Critical power for nonlinearities”*. It remains so in the revised manuscript, and this aspect has been further emphasized in the revised Primary Manuscript.

We also thank Reviewer #3 for noting that the dependence of the effective critical power on the pulse duration is both interesting and well supported by our theoretical approach. As discussed in our reply to Reviewer #3, Point 11, there is currently no available model capable of carrying out rigorous and complete propagation simulations for sub-picosecond and picosecond pulses. In fact, a direct implication of our results is that any model using literature values for the critical power and/or effective multi-photon absorption is inherently flawed for predicting the studied regime. The values determined in our work therefore represent a necessary and substantial step toward realistic modeling of the interaction. The comparison to existing literature values is thus not incidental but central to our manuscript, especially given that for longer pulses the effective nonlinear coefficients deviate even more from literature values that assume pulse-duration independence.

Finally, we also respectfully disagree with Reviewer #3 who does not consider, without any justification, our methodology to be the most appropriate for determining nonlinear coefficients with precision. As we highlight in the manuscript through our comparison with the z-scan technique, our methodology operates under application-relevant laser conditions—tight focusing, high input pulse energy, and the same pulse durations used in practical scenarios. Therefore, we are confident that our method provides realistic and directly applicable nonlinear coefficients.

Revisions brought to the manuscript: The original statement:

“We exploit the reduced energy deposition with ultrashort pulses below the modification threshold to determine the three-dimensional (3D) fluence distribution in these materials, which eventually leads us to define key nonlinear interaction parameters, including the peak fluence F_p , the effective critical power $P_{\text{cr}}^{\text{eff}}$, the effective 2- and 3-photon absorption coefficients β_2^{eff} and β_3^{eff} , the fraction of absorbed energy f_E , and the characteristic absorption length L_{abs} .”

has been revised to:

*“We exploit the reduced energy deposition with ultrashort pulses below the modification threshold to determine the three-dimensional (3D) fluence distribution in these materials, which eventually leads us to define key nonlinear interaction parameters, including the peak fluence F_p , the effective critical power *for nonlinearities* $P_{\text{cr}}^{\text{eff}}$, *and* the effective 2- and 3-photon absorption coefficients β_2^{eff} and β_3^{eff} .”* (Primary Manuscript, page 2)

The original statement:

“Such a temporal dependence of the effective critical power was observed in air (...)”

has been revised to:

*“Such a temporal dependence of the effective critical power was observed *for ultrashort pulse propagation in air* (...)”* (Primary Manuscript, page 3)

The original statement:

“The corresponding calculations serve as a benchmark for the evaluation of key interaction parameters such as the effective critical power P_{cr}^{eff} (see Supplementary Information, Section S4.3), which is in turn implemented in our model based on a modified Marburger formula to determine the effective 2- and 3-photon absorption coefficients β_2^{eff} and β_3^{eff} (see Supplementary Information, Section S4.4).”

has been revised to:

“The corresponding calculations serve as a benchmark for the evaluation of key interaction parameters such as the effective critical power **for nonlinearities** P_{cr}^{eff} (see Supplementary Information, Section S4.3), which is in turn implemented in our model based on a modified Marburger formula to determine the effective 2- and 3-photon absorption coefficients β_2^{eff} and β_3^{eff} (see Supplementary Information, Section S4.4).” (Primary Manuscript, Materials & methods)

To better showcase these results, Fig. 3 in the originally manuscript has been split into two figures. In the revised manuscript, Fig. 3 shows the dependence of the nonlinear optical coefficients as a function of the pulse duration, and Fig. 4 shows transversely space-integrated measurements to determine the fraction of absorbed energy, and the characteristic absorption length.

Reviewer #3, Point 13

Reviewer #3: The influence of pulse duration and wavelength is confirmed, but only two wavelengths are used (1.5 and 1.9 μm), which are relatively close. Thus, the conclusions about mid-IR and high-order multiphoton regimes are rather speculative.

Authors: By selecting two different wavelengths, our intention was not to perform a full parametric study on the influence of wavelength on filamentation in semiconductors, but rather to specifically explore the effect of the multi-photon absorption (MPA) order. Although the wavelengths $\lambda = 1555$ nm and $\lambda = 1960$ nm are relatively close in absolute value, they correspond to different MPA regimes for InP and GaAs— $N = 2$ at 1555 nm and $N = 3$ at 1960 nm. This change in absorption order strongly modifies the ionization dynamics and the resulting energy deposition. We found that increasing the MPA order from 2 to 3 is beneficial for optimizing energy deposition—with more than one order of magnitude increase in the peak fluence F_p observed when using $N = 3$ instead of $N = 2$. This increase is consistent with results obtained in silicon at $\lambda = 4.6$ μm ($N = 5$), where energy deposition is sufficient to produce permanent modification (see Ref. [31] and the related discussion in Reviewer #3, Point 3).

Reviewer #3, Point 14

Reviewer #3: The sensitivity to chirp is an interesting observation. However, it seems to show a dependence, rather than demonstrate a major “control parameter”.

Moreover, with rather narrow spectrum (< 30 nm), it is hard to attribute this to change in nonlinear effects without invoking resonance effects. Also, in experiments, any chirp change is usually associated with variation in pulse temporal shape and asymmetry, which can also affect results. Thus, definitive conclusions should be avoided unless complete temporal and spectral characterization is provided and/or results can be supported by rigorous simulations (see above).

Authors: We thank Reviewer #3 for noting that “the sensitivity to chirp is an interesting observation”. As shown in our study, an increase in the peak fluence F_p in Si by a factor of 2.4 is observed for the same pulse duration of $\tau = 3$ ps [see Primary Manuscript, Fig. 5(b)]. This effect is indeed less pronounced than the variations observed when the pulse duration or the multi-photon absorption order are increased [see Primary Manuscript, Figs. 3(a) and 5(e), respectively]. Nevertheless, this factor of 2.4 is significant for improving energy deposition.

Reviewer #3 raises the possibility of resonance effects. However, for our central wavelength of $\lambda = 1960$ nm in Si, no resonant multiphoton condition exists between the 2PA and 3PA thresholds (see discussion in Reviewer #3, Point 7). Therefore, the observed chirp dependence cannot originate from a resonance effect.

Reviewer #3 also notes that chirp changes can modify the temporal pulse shape and asymmetry. This is indeed true for spectra with strong modulations. In our case, the measured input spectrum is smooth, as confirmed by new high-resolution spectral characterizations now added in the Supplementary Information. Fourier-transform

calculations show that the temporal intensity profiles are nearly identical for up- and down-chirped pulses. Moreover, in order to ensure that the small discrepancies between the temporal intensity profiles calculated for the two chirps are negligible, we have verified that they at least do not affect ionization under the simplest assumptions. Accounting for two-photon ionization, the temporal evolution of the electron density n_e is governed by

$$\frac{\partial n_e}{\partial t} = \frac{\beta_2}{2\hbar\omega} \left(1 - \frac{n_e}{n_{\text{at}}}\right) I^2, \quad (\text{R5})$$

where $\beta_2 = 1.38 \times 10^{-12}$ m/W is the two-photon photon absorption coefficient taken from Ref. [R14] for a wavelength of $\lambda = 1960$ nm, \hbar is the reduced Planck constant, $\omega = 2\pi c/\lambda$ is the central angular frequency, $n_{\text{at}} = 5 \times 10^{22}$ cm $^{-3}$ is the atomic density, and I is the laser intensity. The initial electron density is assumed to be on the order of 10^{10} cm $^{-3}$, which is a typical value for intrinsic Si. We calculated the temporal evolution of n_e for up- and down-chirped pulses. As shown in Fig. R1, the influence of the chirp on the evolution of the electron density produced by two-photon absorption is negligible—especially for high n_e values.

Therefore, the aforementioned factor of 2.4 on the peak fluence F_p between up- and down-chirped cannot be caused by variations in pulse temporal shape and asymmetry between the two chirp configurations. This strongly supports our main conclusion that the observed effect of the chirped is the caused by a change in the ionization dynamics.

Fig. R1 Temporal evolution of the laser intensity (dashed lines, left vertical axis), and the electron density n_e (solid lines, right vertical axis) for up- and down-chirped pulses.

Revisions brought to the manuscript: The new Section titled “S3.4 Chirped temporal profiles” has been added to the Supplementary Information. This Section includes the new Fig. S8 showing fine characterizations of the input laser spectrum as well as calculations of the temporal intensity profile in the up- and down-chirp configurations, compared to bandwidth-limited pulses.

The original statement:

“The chirp-dependence of filamentation may originate mainly from two physical phenomena. First, the ionization dynamics could differ between up- and down-chirped pulse. Calculations showed that a similar asymmetry exists for fused silica and MgF₂ [44, 45, 47]. This is ascribable to the increased efficiency of multi-photon absorption for shorter wavelengths (even for the same multi-photon absorption order [48, 49]), while avalanche ionization is more efficient for longer wavelengths. Therefore, in the down-chirped configuration where the “blue” spectral components arrive before the “red” ones, the produced free-carrier density is higher. The second phenomenon which could play a role is nonlinear dispersion induced by the plasma. Indeed, the interaction results in extended plasma channels along the optical axis, which may show anomalous dispersion, meaning that the “blue” can catch up with the “red” part of the pulse, in turn resulting in shorter duration—and thus, higher intensity.”

has been revised to:

“Given that, for the measured smooth input spectrum, changing the chirp from up to down changes insignificantly the temporal intensity profile (see Supplementary Information, Section S3.4), the higher peak fluence observed for down-chirped pulses can be explained by differences in ionization dynamics. Numerical studies for materials such as fused silica and MgF₂ [44, 45, 47] showed a similar asymmetry, as a consequence of multi-photon absorption being more efficient at shorter wavelengths—even for the same multi-photon order, as shown

in Si [48, 49]—whereas avalanche ionization becomes more efficient at longer wavelengths. The latter trend is directly described by the Drude model, which predicts that the inverse Bremsstrahlung heating rate scales as λ^2 , so free carriers gain energy more efficiently from longer-wavelength fields. In the down-chirped configuration, the blue spectral components (shorter wavelengths) arrive earlier in the pulse, producing a higher initial free-carrier density that enhances avalanche ionization by the subsequent red components (longer wavelengths). As a secondary mechanism, plasma-induced anomalous dispersion may further strengthen this effect by causing the blue components to catch up with the red ones, compressing the pulse and increasing its peak intensity.”
(Primary Manuscript, page 5)

Reviewer #3, Point 15

Reviewer #3: Fig. 1 compiles refractive index data from literature for many semiconductors and dielectrics which is surely interesting but appear not to be essential for the discussions of the paper.

Authors: We agree with Reviewer #3 that the linear refractive index data are not essential for the discussions in the manuscript.

Revisions brought to the manuscript: In the Primary Manuscript, Fig. 1 now shows only the evolution of nonlinear refractive index n_2 as a function of the band gap E_g . The linear refractive index values n_0 are still conserved in the Supplementary Information, Section S1.

References

- [R1] Mouskeftaras, A. *et al.* Self-limited underdense microplasmas in bulk silicon induced by ultrashort laser pulses. *Applied Physics Letters* **105**, 191103 (2014). URL <https://doi.org/10.1063/1.4901528>.
- [R2] Couairon, A., Sudrie, L., Franco, M., Prade, B. & Mysyrowicz, A. Filamentation and damage in fused silica induced by tightly focused femtosecond laser pulses. *Physical Review B* **71**, 125435 (2005). URL <https://doi.org/10.1103/PhysRevB.71.125435>.
- [R3] Zavedeev, E. V., Kononenko, V. V. & Konov, V. I. Delocalization of femtosecond laser radiation in crystalline Si in the mid-IR range. *Laser Physics* **26**, 016101 (2016). URL <https://doi.org/10.1088/1054-660X/26/1/016101>.
- [R4] Chichkov, B. N., Momma, C., Nolte, S., Alvensleben, F. & Tünnermann, A. Femtosecond, picosecond and nanosecond laser ablation of solids. *Applied Physics A* **63**, 109–115 (1996). URL <https://doi.org/10.1007/BF01567637>.
- [R5] Gattass, R. R. & Mazur, E. Femtosecond laser micromachining in transparent materials. *Nature Photonics* **2**, 219–225 (2008). URL <https://doi.org/10.1038/nphoton.2008.47>.
- [R6] Yin, Z. *et al.* Experimental quantum-enhanced kernel-based machine learning on a photonic processor. *Nature Photonics* (2025). URL <https://doi.org/10.1038/s41566-025-01682-5>.
- [R7] Chambonneau, M. *et al.* Positive- and negative-tone structuring of crystalline silicon by laser-assisted chemical etching. *Optics Letters* **44** (2019). URL <https://doi.org/10.1364/OL.44.001619>.
- [R8] Saltik, A. & Tokel, O. Laser-written wave plates inside the silicon enabled by stress-induced birefringence. *Optics Letters* **49**, 49 (2024). URL <https://doi.org/10.1364/OL.504600>.
- [R9] Zhang, J. *et al.* Femtosecond laser filament ablated grooves of SiC ceramic matrix composite and its grooving monitoring by plasma fluorescence. *Ceramics International* **50**, 16474–16480 (2024). URL <https://doi.org/10.1016/j.ceramint.2024.02.133>.
- [R10] Salimullah, M. & Tripathi, V. K. Filamentation of laser radiation in DC biased GaAs. *Applied Physics* **22**, 89–93 (1980). URL <https://doi.org/10.1007/BF00897938>.
- [R11] Strickland, D. & Mourou, G. Compression of amplified chirped optical pulses. *Optics Communications* **55**, 447–449 (1985). URL [https://doi.org/10.1016/0030-4018\(85\)90151-8](https://doi.org/10.1016/0030-4018(85)90151-8).
- [R12] Sudrie, L., Franco, M., Prade, B. & Mysyrowicz, A. Study of damage in fused silica induced by ultra-short IR laser pulses. *Optics Communications* **191**, 333–339 (2001). URL [https://doi.org/10.1016/S0030-4018\(01\)01152-X](https://doi.org/10.1016/S0030-4018(01)01152-X).
- [R13] Rahnama, A., Mahmoud Aghdami, K., Kim, Y. H. & Herman, P. R. Ultracompact Lens-Less “Spectrometer in Fiber” Based on Chirped Filament-Array Gratings. *Advanced Photonics Research* **1**, 2000026 (2020). URL <https://doi.org/10.1002/adpr.202000026>.
- [R14] Lin, Q. *et al.* Dispersion of silicon nonlinearities in the near infrared region. *Applied Physics Letters* **91**, 021111 (2007). URL <https://doi.org/10.1063/1.2750523>.
- [R15] Fedorov, V. Y., Chanal, M., Grojo, D. & Tzortzakis, S. Accessing Extreme Spatiotemporal Localization of High-Power Laser Radiation through Transformation Optics and Scalar Wave Equations. *Physical Review Letters* **117**, 043902 (2016). URL <https://doi.org/10.1103/PhysRevLett.117.043902>.
- [R16] Chanal, M. *et al.* Crossing the threshold of ultrafast laser writing in bulk silicon. *Nature Communications* **8**, 773 (2017). URL <https://doi.org/10.1038/s41467-017-00907-8>.
- [R17] Mareev, E. I. *et al.* Effect of pulse duration on the energy delivery under nonlinear propagation of tightly focused Cr:forsterite laser radiation in bulk silicon. *Laser Physics Letters* **17**, 015402 (2020). URL <https://doi.org/10.1088/1612-202X/ab5d23>.
- [R18] Swain, A. B., Kuttruff, J., Vorberger, J. & Baum, P. Stronger femtosecond excitation causes slower electron-phonon coupling in silicon. *Physical Review Research* **7**, 023114 (2025). URL <https://doi.org/10.1103/PhysRevResearch.7.023114>.
- [R19] Kilen, I. *et al.* Propagation Induced Dephasing in Semiconductor High-Harmonic Generation. *Physical Review Letters* **125**, 083901 (2020). URL <https://doi.org/10.1103/PhysRevLett.125.083901>.
- [R20] Tsaturyan, A., Kachan, E., Stoian, R. & Colombier, J.-P. Ultrafast bandgap narrowing and cohesion loss of photoexcited fused silica. *The Journal of Chemical Physics* **156** (2022). URL <https://doi.org/10.1063/5.0096530>.
- [R21] Kachan, E., Tsaturyan, A., Stoian, R. & Colombier, J.-P. First-principles study of ultrafast bandgap dynamics in laser-excited α -quartz. *The European Physical Journal Special Topics* **232**, 2241–2245 (2023). URL <https://doi.org/10.1140/epjs/s11734-022-00747-8>.

Second response to Reviewers

Contents

Response to Reviewer #1	2
Reviewer #1, Recommendation	2
Response to Reviewer #3	3
Reviewer #3, Foreword	3
Reviewer #3, Point 1	3
Reviewer #3, Point 2	4
Reviewer #3, Point 3	5
References	7

Response to Reviewer #1

Reviewer #1, Recommendation

Reviewer #1: The authors have addressed all issues that were pointed out in my review. Therefore, I recommend the paper for publication.

Authors: We thank Reviewer #1 for their positive evaluation and recommendation for publication.

Response to Reviewer #3

Reviewer #3, Foreword

Reviewer #3: As indicated in my original report, the paper by Chambonneau et al. addresses important scientific and engineering questions related to the development of new technologies based on in-volume interactions in semiconductors. In particular, the experimental work is carried out in configurations that are especially relevant to the challenge of 3D writing.

The revisions have produced a version that is convincing in some respects, notably through clearer definitions of the manipulated quantities (e.g., maximum fluence) and improved use of terminology. However, it remains less convincing with regard to the major conceptual concerns raised in the first round of review, as discussed below. Moreover, the additional material introduced in response to reviewers' comments, although informative in itself, tends at times to overload an already dense manuscript without significantly enhancing the clarity or impact of the conclusions.

Overall, I again recognize the substantial experimental effort and the merit of compiling such a large body of measurements. Nevertheless, I must reaffirm my initial impression: the work does not present a sufficiently novel conceptual contribution, nor does it offer a decisive advance in understanding or a clear technological direction. I therefore believe that the manuscript would be more appropriate for a more specialized journal.

In the present report, I do not revisit all points of disagreement with the authors but focus on a few key concerns that remain and that justify my overall assessment.

Authors: We thank Reviewer #3 for re-evaluating our work and for acknowledging the relevance of the configuration and the scope of the dataset. For clarity, reviewer comments are shown in **blue**, and our responses in **green**. No further manuscript changes were introduced in this resubmission.

Reviewer #3, Point 1

Reviewer #3: First, although I continue to question the appropriateness of grouping all observed effects under the term “filamentation,” I note and welcome the removal of the earlier claim of “universality,” which was not well supported and appeared overstated. Unfortunately, this change has been accompanied by the new claim of “Extreme optical nonlinearities,” which I find again somewhat artificial and even confusing once one considers how these values are obtained. The authors appropriately refer to “effective” nonlinear coefficients, but the extreme values they report arise primarily from the use of simplified retrieval models. While these effective coefficients are of interest for the particular experimental conditions studied here, they do not contradict the literature values obtained by standard methods and cannot be easily extrapolated to other conditions. Consequently, I do not consider it reasonable to use them to draw quantitative conclusions or make claims of broad applicability to different configurations such as looser focusing or waveguide-based applications (THz generation, high harmonics, supercontinuum, etc.).

Authors:

On the determination of “extreme” effective parameters. The reviewer states that the “*extreme values*” reported in our manuscript “*arise primarily from the use of simplified retrieval models*”.

The effective critical power is obtained directly from experiment—specifically from the measured onset of nonlinearities—**not from any retrieval model**. The effective multiphoton absorption coefficient, obtained using the Marburger formula (Ref. 41), **is further refined by explicitly accounting for nonlinear losses**—one of the key innovations of our work. Notably, we show in Supplementary Information, Fig. S15 that this theoretical refinement is essential to realistically model in-volume laser–semiconductor interaction.

On the comparison with the literature. The claim that our results “*do not contradict the literature values obtained by standard methods*” is not supported by the data.

Figures 3(b) and 3(c) show deviations—up to orders of magnitude—from the small-signal dashed-line literature values. The reported quantities are experimentally deduced effective coefficients at the propagation scale: they encode the net influence of nonlinear refraction and absorption, including both instantaneous and delayed carrier dynamics, in the strongly driven regime. These effective coefficients influence—but do not replace—separate mechanisms such as plasma-induced defocusing. Consequently, they cannot be mapped one-to-one onto small-signal constants; equating them conflates parameters with different definitions and domains.

On the extrapolation to other applications. Finally, Reviewer #3 suggests that the determined nonlinear coefficients are valid only under “*particular conditions*” and cannot be “*extrapolated to other configurations such as looser focusing or waveguide-based applications (THz generation, high harmonics, supercontinuum, etc.)*”. This criticism ignores our detailed response to Reviewer #3, Point 9 in the previous round of review, where we recalled that Ref. 24 **already demonstrated eight years ago** that similar features are observed under different focusing conditions. Moreover, in our study, the parameters were varied so that the input laser intensity spanned eight orders of magnitude (10^6 – 10^{14} W/cm²), thereby **supporting applicability across the reported vast parameter space**. In addition, the reviewer overlooks our new experiments on supercontinuum generation (performed in response to Reviewer #1, Point 11), which directly evidence that our findings are not limited to ultrafast laser writing but **extend to other nonlinear propagation regimes and applications**.

Reviewer #3, Point 2

Reviewer #3: Regarding the chirp dependence, I again acknowledge the interesting observation of a sensitivity to chirp, but I do not find in the added material more convincing evidence that chirp can be considered as a critical control parameter. In particular, the data do not demonstrate a sufficiently broad spectral coverage to fully support the discussions. Also, the observed symmetry of the temporal profiles appears to result from the reconstruction procedure itself, which retains only the quadratic phase term in Eq. S10 without measurement.

Authors:

On sufficiently broad spectrum. Reviewer #3 claims that “*the data do not demonstrate a sufficiently broad spectral coverage to fully support the discussions*”.

We appreciate the reviewer’s interest in the chirp dependence. However, the claim that our spectral coverage is insufficient is unsupported by any quantitative criterion, making this statement subjective.

Moreover, this comment contradicts previous studies that reported analogous chirp-dependent effects in dielectrics using much narrower bandwidths: 7.5 nm (Ref. 44), 20.9 nm (Ref. 45), and 3.9 nm for chirp-dependent laser particle acceleration (Ref. 43). All of these are **significantly smaller than the 29 nm bandwidth** employed in our study. Quantitatively, the two-photon absorption coefficient for the red side of the employed spectrum is 10.5% lower than for the blue side [for Si, see Fig. 3(a) in Ref. 49], while free-carrier absorption scales as λ^2 , making it 3% stronger for longer wavelengths. These considerations **strongly support the difference we observe in F_p between up- and down-chirped pulses** in Figs. 5(a)–5(c).

On temporal–spectral characterizations. Regarding the temporal-spectral characterization, Reviewer #3 considers that “*the observed symmetry of the temporal profiles appears to result from the reconstruction procedure itself, which retains only the quadratic phase term in Eq. S10*”.

By definition, **the quadratic phase term dominates** pulse-duration variation when tuning the compressor in chirped pulse amplification. The reviewer omits that the phase standardly defined in Eq. (S10) is written as a Taylor expansion, meaning that **higher-order terms are negligible** compared to the quadratic term. Furthermore, the observed symmetry of the temporal profile does not result from the reconstruction: as shown in Fig. S8, **it primarily originates from the symmetry of the measured spectrum**.

To prove this, let us consider the hypothetical asymmetric spectrum displayed in Fig. R2-1(a). By using the same reconstruction method, the temporal intensity profile is calculated in Fig. R2-1(b) for different second-order dispersion coefficients b_2 . In the case of a bandwidth-limited pulse (red, $b_2 = 0$), the temporal intensity profile is symmetric. This is to be expected since all spectral components are in phase, and the asymmetry in the spectrum is not transferred to the time domain. However, in the case of a chirped pulse identical as in Supplementary Information, Fig. S8 [green, $b_2 = -(3 \times 10^{-12})^2$ s²], the asymmetry of the spectrum is also contained in the temporal intensity profile. This shows that the observed symmetry does not “*result from the reconstruction procedure itself*” as stated by Reviewer #3, but is in fact **a hallmark of the symmetric spectrum used in the experiments**.

Fig. R2-1 (a) Hypothetical asymmetric spectrum. (b) Corresponding temporal intensity profiles calculated assuming a bandwidth-limited pulse [red, $b_2 = 0$], and a chirped pulse [green, $b_2 = -(3 \times 10^{-12})^2 \text{ s}^2$].

Reviewer #3, Point 3

Reviewer #3: Finally, a major concern raised by reviewers in the first round was related to the absence of a meaningful quantitative metric to evaluate progress in addressing the challenges of 3D writing in semiconductors under nonlinear propagation effects. The metric introduced in the revision, the fluence F_p , seems to me entirely inappropriate. What matters here is the absorbed energy density (or maybe ionization level), not the maximum achievable fluence inside the material. To make this error obvious, one can consider the limiting case of a continuous, low-intensity radiation in the transparency domain of a semiconductor: in such a situation F_p can become arbitrarily large while the nonlinear absorption—and thus energy deposition—tends toward zero. This conceptual inconsistency undermines the reported ratios of improvement and the hierarchy of parameters (pp. 6–7). I would in fact strongly recommend suppressing these new sections, as they are misleading and add confusion rather than clarity.

Authors:

On the relevance of F_p metric. First, Reviewer #3 asserts that “the metric introduced in the revision, the fluence F_p , seems entirely inappropriate” as “a meaningful quantitative metric to evaluate progress in addressing the challenges of 3D writing”.

The fluence F_p was already present in the originally submitted manuscript (NCOMMS-25-38349-T); implying that it was newly introduced is therefore misleading. Moreover, the reviewer narrows the scope of our work to 3D laser writing, again ignoring the broader relevance we demonstrated (including supercontinuum generation).

On “what matters” (absorbed energy density) versus what can be measured. Reviewer #3 contends that “what matters is the absorbed energy density”.

However, the determination of the peak fluence F_p in Si using nonlinear propagation imaging for a given pulse duration has been well established in prior high-impact publications—see, for instance, Fig. 4 in Ref. 17 (Physical Review Letters), and Figs. 1(b) and Fig. 2 in Ref. 24 (Nature Communications). In Fig. 4 of our manuscript, we further introduced the fraction of absorbed energy and the characteristic absorption length, which complement F_p in Figs. 2(d), 3(a), 5(b), and 5(e), and provide a quantitative and comprehensive description of energy deposition.

To the best of our knowledge, there is no direct, non-perturbative method to measure the time-dependent optical field $E(t)$ or intensity $I(t)$ inside the bulk of a solid for ultrashort pulses. In practice, one measures the ultrafast response of the material with a delayed probe—e.g., Kerr-induced bire-

fringe, transient transmission or phase, free-carrier density, stress, or luminescence—and infers $I(t)$ using calibrated models (instantaneous Kerr and carrier dynamics). Achieving a truly direct measurement would require local access, at each voxel, to the complex spatio-temporal field, which is not experimentally accessible *in situ* (in particular, autocorrelation alone is non-unique). Consequently, **the most robust bulk-accessible per-pulse metric to report is the peak fluence F_p** , while time-resolved proxies can be obtained via pump-probe imaging and related techniques.

On the “continuous-wave limit” counterexample. The example given by Reviewer #3—considering “*a continuous, low-intensity radiation in the transparency domain of a semiconductor*” as evidence that F_p is meaningless—is irrelevant in the context of ultrafast pulses. Moreover, the argument that “ *F_p can become arbitrarily large while nonlinear absorption tends toward zero*” is physically incorrect. Even for continuous irradiation, nonlinear absorption can take place. In fact, sufficient energy deposition to induce material modification can occur when the deposited energy exceeds the latent heat of fusion. To illustrate this latter aspect, continuous laser irradiation at 1.9 μm has been shown to enable laser welding of silicon [R2-1]. Moreover, continuous light is known to be a powerful tool for the inscription of fiber Bragg gratings in transparent materials [R2-2, R2-3]. Finally, damage induced by continuous laser radiation is a critical issue in large facilities such as the Laser Interferometer Gravitational-Wave Observatory (LIGO) [R2-4].

On removing new paragraphs. Based on this example of continuous radiation that supposedly shows that F_p is inappropriate, the reviewer finally “*recommends suppressing*” the paragraphs on the “*reported ratios of improvement and the hierarchy of parameters (pp. 6–7)*”. These paragraphs were added thanks to Reviewer #1, Point 8 in the previous round of review, and represent a guideline for the readers to select the most appropriate regime with respect to their applications. Given the incorrectness of the example of continuous radiation, we therefore retain these sections, which serve as practical guidance for selecting operating windows.

References

- [R2-1] Sari, F., Hoffmann, W.-M., Haberstroh, E. & Poprawe, R. Applications of laser transmission processes for the joining of plastics, silicon and glass micro parts. *Microsystem Technologies* **14**, 1879–1886 (2008). URL <https://doi.org/10.1007/s00542-008-0675-3>.
- [R2-2] Bennion, I., Williams, J. A., Zhang, L., Sugden, K. & Doran, N. J. UV-written in-fibre Bragg gratings. *Optical and Quantum Electronics* **28**, 93–135 (1996). URL <https://doi.org/10.1007/BF00278281>.
- [R2-3] Dobb, H. *et al.* Continuous wave ultraviolet light-induced fiber Bragg gratings in few- and single-mode microstructured polymer optical fibers. *Optics Letters* **30**, 3296 (2005). URL <https://doi.org/10.1364/OL.30.003296>.
- [R2-4] Gushwa, K. E. & Torrie, C. I. Coming clean: understanding and mitigating optical contamination and laser induced damage in advanced LIGO. In Exarhos, G. J., Gruzdev, V. E., Menapace, J. A., Ristau, D. & Soileau, M. (eds.) *Proceedings of SPIE*, vol. 9237, 923702 (2014). URL <https://doi.org/10.1117/12.2066909>.

Third response to Reviewers

Contents

Response to Reviewer #3	2
Reviewer #3, Foreword	2
Reviewer #3, Point 1	2
Reviewer #3, Point 2	3
Reviewer #3, Point 3	4

Response to Reviewer #3

Reviewer #3, Foreword

Reviewer #3: As a preliminary remark, I observe that the authors' reply does not reassess the conceptual framing of the manuscript, nor does it provide additional experimental evidence that would reinforce the claim of wide applicability. The authors predominantly expand on their disagreements with the comments, yet do not seem to have considered a deeper or more comprehensive re-evaluation of the underlying issues.

Although the response concentrates on the few main points raised by previous report, none convincingly affects my overall evaluation. Again, my position does not concern the intrinsic interest of the study or the merit of the dataset, but reflects instead the high-level standards of originality and impact expected for this journal. From this perspective, the breadth of impact and the significance of the advances presented remain, in my view, insufficiently substantiated.

Regarding author response to R3 – Foreword I appreciate the authors' efforts in providing a point-by-point reply. However, I note that the general concerns previously raised regarding conceptual novelty and/or understanding remain insufficiently addressed. The response focuses on individual remarks on the selection of deficiencies but does not directly engage with the broader issues which have been expressed.

Authors: We thank Reviewer #3 for the continued engagement and for clearly restating the remaining concerns regarding the conceptual framing and the scope of applicability. Since the manuscript has already been revised substantially in the previous rounds (including clearer definitions and terminology), we do not introduce further manuscript changes in this resubmission; instead, we address the reviewer's overarching point as directly as possible by clarifying here the intended scope of the work as it is stated in the manuscript. Our contribution is the experimental determination of effective parameters and scaling trends in the investigated regime (mid-IR excitation, ultrashort pulses, and tight focusing in bulk semiconductors), where Kerr self-focusing and plasma-driven effects jointly govern the interaction. These effective parameters are defined at the propagation scale (i.e., effective coefficients that reproduce the observed self-focusing/onset and nonlinear-loss behavior over the propagation path in our regime), and are intended for quantitative description and modeling of strongly driven nonlinear propagation (rather than as a replacement for small-signal material constants); within this framework, they enable comparison across materials and operating conditions and provide practical inputs for predicting energy-deposition trends in related nonlinear propagation scenarios. For clarity, reviewer comments are shown in **blue**, and our responses in **green**.

Reviewer #3, Point 1

Reviewer #3: Regarding author response to R3 – P1

I maintain that the Marburger formula, even when refined to account for nonlinear losses, while effective in describing the observations under the specific conditions of this study, cannot substitute to a rigorous nonlinear propagation model to support fully quantitative conclusions. Without a clear demonstration of the validity of the reported nonlinear coefficients in more general scenarios, they cannot be considered as contradicting others found in the literature.

Although the authors point out that supercontinuum generation is shown in S17, these observations remain relatively qualitative and do not directly validate the measured nonlinear coefficients. To substantiate claims of broad applicability, one would reasonably expect a range of different experiments (e.g., SC, THz, harmonic generation, material modification,) combined/compared with modeling that consistently supports the general relevance of the reported coefficients.

Authors:

On our developed theoretical approach compared to a rigorous nonlinear propagation model. Reviewer #3 maintains that the Marburger formula, even when refined to account for nonlinear losses, cannot substitute for a fully rigorous nonlinear propagation model “to support fully quantitative conclusions”. We agree that our approach is not intended to replace a full spatio-temporal propagation simulation; rather, it provides an experimentally anchored route to determine propagation-scale effective coefficients in a regime where a comprehensive, predictive model is currently not available across the full range of pulse durations addressed here. As indicated in our first response (Reviewer #3, Point 11), this is especially true for pulses > 1 ps where carrier and lattice dynamics evolve during the pulse and strongly couple to propagation. A rigorous 3D model would require self-consistent treatment of propagation, ionization and transport, carrier–phonon energy exchange,

temperature-dependent material properties (e.g., band-gap renormalization), and potentially hydrodynamics, making it intrinsically multi-scale and computationally demanding. The goal of the present work is therefore to establish the experimentally relevant nonlinear coefficients and their temporal scaling laws that constitute necessary inputs and benchmarks for such future comprehensive models. In this context, the substantial deviations we report from small-signal literature values are not a matter of semantics: they reflect strong-field, propagation-scale effective behavior that is directly relevant to quantitative modeling in the investigated regime.

On the claim of broad applicability. Reviewer #3 suggests that substantiating broad applicability would require a wide set of additional experiments (e.g., SC, THz, HHG, material modification) combined with modeling that consistently supports the reported coefficients. We respectfully note that our claim of applicability is tied to the propagation regime and parameter space investigated in this study, and is supported by (i) cross-material consistency of the observed propagation trends, (ii) systematic variations of pulse duration, chirp, and wavelength, and (iii) the additional supercontinuum measurements included in the Supplementary Information. Rather than presenting a claim of exhaustive validation across all nonlinear-optics platforms, we provide a quantitative reference set of effective coefficients and scaling laws that can be directly used as inputs and benchmarks for modeling and for designing experiments in closely related nonlinear-propagation scenarios. As these coefficients enter the description of a broad range of strong-field phenomena, we outline representative connections to supercontinuum generation, THz generation, high-harmonic generation, and laser modification in Section 4.6 of the Supplementary Information, while keeping the present work focused on the experimentally validated propagation-scale determination of the parameters.

Reviewer #3, Point 2

Reviewer #3: Regarding author response to R3 – P2

The authors' explanation regarding spectral dependencies offers some clarity that could improve the paper. Accordingly, I agree that even a relatively narrow spectrum may potentially lead to noticeable variations. However, the question of potential pulse asymmetry remains insufficiently addressed. While the quadratic phase term is indeed an appropriate simplification to describe the effect introduced by the stretcher, my initial remark was related to the spectral phase prior to the stretcher. Assuming an initially flat spectral phase leads naturally to a symmetric amplitude profile, but this assumption is not demonstrated. Figure R2-1 does not help to provide an unambiguous confirmation of symmetry in the conditions studied, and additional characterization would be required to fully support this point.

Authors:

On potential pulse asymmetry. We thank Reviewer #3 for recognizing that the explanation we gave in the second response “*offers some clarity*”, and for agreeing that the spectrum we used potentially leads to “*noticeable variations*”. Reviewer #3 also clarified their original comment concerning potential pulse asymmetry, which could be related to a non-flat spectral phase prior to the stretcher.

We emphasize that we do not assume an initially flat spectral phase, nor do we claim that the temporal intensity profile is perfectly symmetric. We agree that a full retrieval of the spectral phase (e.g., by FROG/SPIDER) would be required to uniquely reconstruct the exact field; however, our conclusions regarding chirp effects do not rely on perfect temporal symmetry. In particular, the physical interpretation discussed in the manuscript is driven by the temporal ordering of spectral components (up- vs down-chirp) combined with wavelength-dependent ionization/heating mechanisms, rather than by fine details of the temporal envelope. Moreover, any residual higher-order spectral phase present upstream would be common to both chirp settings, and therefore cannot reasonably account for the systematic difference observed between up- and down-chirped pulses.

To directly assess the sensitivity to possible asymmetry, we performed a conservative test within the same rate-equation framework used in our first response to Reviewer #3, Point 14. The temporal evolution of the electron density is calculated in Fig. R3-1 for a symmetric 3-ps Gaussian pulse (red) and for an asymmetric 3-ps pulse (blue) obtained by applying a quadratic spectral phase (yielding 3 ps) to the experimentally measured spectral amplitude and Fourier transforming (thereby incorporating small spectral-amplitude modulations into the temporal profile). As expected, small modulations in the power spectral density yield small modulations of the temporal intensity profile (dashed blue vs dashed red). Nevertheless, these modulations translate into a negligible change of the ionization dynamics: the resulting electron-density curves (solid lines) are nearly identical, and the ratio between the maximum electron densities produced by the two pulses differs by less than 1%. We therefore conclude that a plausible degree of pulse-shape asymmetry cannot significantly affect the carrier generation dynamics in our conditions and does not alter the interpretation of the observed chirp

dependence.

Fig. R3-1 Temporal evolution of the laser intensity (dashed lines, left vertical axis), and the electron density n_e (solid lines, right vertical axis) for 3-ps symmetric (red) and asymmetric (blue) pulses. The pulse energy is $E_{in} = 100$ nJ, and the beam radius at $1/e^2$ is $w_0 = 2.2$ μm .

Reviewer #3, Point 3

Reviewer #3: Regarding author response to R3 – P3

The response contains a certain inconsistency: the authors state that the relevance of the chosen parameters extends well beyond 3D writing, yet they justify their choice primarily by referring to studies on 3D writing or material modification. While I agree that using a directly measurable metric is advantageous, my concerns remain regarding its general relevance for a broad range of nonlinear optics experiments.

The issue I initially raised with a metric that can become arbitrarily large even in a regime insufficient to trigger any nonlinear response has not been convincingly addressed. Referring to works involving CW-induced modifications does not clarify how the proposed metric can be consistently used to identify an optimum in the present context. Therefore, I must respectfully but firmly disagree with the authors' assertion that my earlier remarks were "incorrect" and I must maintain my position regarding the limited relevance of the reported ratios.

Authors:

On the primary focus on laser direct writing. In our manuscript (Primary Manuscript, abstract and introduction; Supplementary Information, Section 4.6), we mention multiple potential applications where our work could constitute a critical step. These applications include not only laser direct writing, but also backside processing, microelectronics security, THz wave generation, high-harmonic generation, and supercontinuum generation. Reviewer #3 rightly recalls that the primary focus is on laser direct writing. The main reason is that this is one of the most extreme case where light intensity is sufficiently high to permanently modify semiconductors internally. However, our results are not restricted to the "writing" end-point: the reported propagation-scale effective coefficients and their scaling laws are extracted from nonlinear propagation itself (self-focusing, nonlinear losses, and plasma-driven effects) over a wide range of input intensities and pulse durations, and therefore directly inform other strong-field nonlinear-propagation scenarios in bulk semiconductors. In this sense, direct writing serves as a motivating and technologically relevant benchmark, but not as the sole justification for the extracted parameters.

On the metric that can become arbitrarily large. The example previously given by Reviewer #3 was: "a continuous, low-intensity radiation in the transparency domain of a semiconductor: in such a situation F_p can become arbitrarily large while the nonlinear absorption—and thus energy deposition—tends toward zero". We believe that this counterexample does not address the way F_p is defined and used in our study. In our work, F_p is a per-pulse quantity defined from ultrashort-pulse experiments as the saturation (plateau) value of F_{max} when increasing E_{in} , i.e., $F_p = \max_{E_{in}}(F_{max})$ in the nonlinear regime where F_{max} ceases to increase due to nonlinear propagation and losses. In a regime insufficient to trigger nonlinear response (linear propagation / CW-like low intensity), such a saturation plateau does not occur, and therefore the "peak saturation fluence" F_p is not the relevant quantity.

- The notion of “fluence” as used here is a per-pulse energy-per-area quantity. For a continuous radiation ($I = I_0$), one would rather discuss intensity and absorbed power, or define an energy-per-area over a chosen exposure time. In that case, $F = I_0 \times t$ grows with t , but this growth is unrelated to the per-pulse saturation concept captured by F_p .
- Even considering an extremely long laser pulse (for instance, in the millisecond range), the energy deposition does not tend to zero. The pulse parameters would be defined the same way as in our manuscript (pulse energy E_{in} , pulse duration τ and beam radius at $1/e^2$ w_0). The maximum fluence would read $F_{\text{max}} = 2E_{\text{in}}/(\pi w_0^2)$, which is a finite metric. Nonlinear absorption—and thus energy deposition—would depend on the maximum intensity $I_{\text{max}} = F_{\text{max}}/\tau$, which does not tend to zero as the increase in τ will also result in an increase in F_{max} [see Primary Manuscript, Fig. 3(a)].

For these reasons, F_p remains a robust bulk-accessible per-pulse metric in the ultrafast nonlinear-propagation regime investigated here, especially when interpreted together with the absorbed-energy fraction and absorption-length observables reported in our manuscript. We agree that our wording in the previous response was stronger than necessary; our intended point was that the CW limiting case is not a meaningful test of a per-pulse saturation metric.